# HERO: Heterogeneous Continual Graph Learning
# via Meta-Knowledge Distillation

## Abstract

Machine learning on heterogeneous graphs has experienced rapid advancement in recent years, driven by the inherently heterogeneous nature of real-world data. However, existing studies typically assume the graphs to be static, while real-world graphs are continuously expanding. This dynamic nature requires models to adapt to new data while preserving existing knowledge. To this end, this work addresses the challenge of continual learning on heterogeneous graphs by introducing HERO (HEterogeneous continual gRaph learning via meta-knOwledge distillation), a unified framework for continual learning on heterogeneous graphs. HERO employs meta-adaptation, a gradient-based meta-learning strategy that provides directional guidance for rapid adaptation to new tasks with limited samples. To enable efficient and effective knowledge reuse, we propose DiSCo (Diversity Sampling with semantic Consistency), a heterogeneity-aware sampling method that maximizes target node diversity and expands subgraphs along metapaths, retaining key semantic and structural information with minimal overhead. Furthermore, HERO incorporates heterogeneity-aware knowledge distillation, which aligns knowledge at both the node and semantic levels to balance adaptation and retention across tasks. Comprehensive evaluations across four benchmark datasets validate HERO's effectiveness in handling continual learning scenarios on expanding heterogeneous graphs.

## 1. Introduction

In recent years, heterogeneous graphs have emerged as a powerful structure for modeling complex real-world sys-

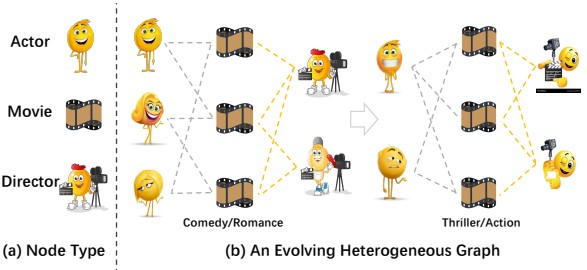

*Figure 1.* An example of Evolving Heterogeneous Graph. (a) Node types include Actor, Movie, and Director. (b) As the graph evolves, new domains emerge (e.g., movies of different genres such as Comedy/Romance and Thriller/Action).

tems across diverse domains, including biological systems, social networks, and recommendation systems (Shi et al., 2018). Unlike homogeneous graphs with uniform node and edge types, heterogeneous graphs capture intricate relationships among multiple node and edge types, enabling more nuanced and comprehensive representations for the target systems (Sun & Han, 2012). For example, in recommendation system research, the user-item networks contain user nodes, item nodes, and the purchase relationship. In a cellular system, the signalling network encodes the complex interaction among different types of entities including small molecules, proteins, and genes (Zitnik et al., 2018). To extract meaningful insights from heterogeneous graphs, different Heterogeneous Graph Neural Networks (HGNNs) have been developed. These approaches mainly fall into two categories: metapath-based HGNNs (Fu et al., 2020; Zhang et al., 2019; Wang et al., 2019; Schlichtkrull et al., 2018; Dong et al., 2017; Sun et al., 2011; Yun et al., 2019; Yang et al., 2023; Zhu et al., 2024), which aggregate information over predefined meta-paths to mine the intricate relationship among the heterogeneous nodes and edges, and meta-path-free models (Hu et al., 2020; Lv et al., 2021; Zhu et al., 2019; Hong et al., 2020; Zhang et al., 2022b; Fu et al., 2023; Huo et al., 2025) that automatically learn node interactions without preconfigured paths, offering enhanced adaptability but potentially sacrificing interpretability.

Despite the success of Heterogeneous Graph Neural Networks (HGNNs) in static settings, real-world graphs are inherently evolving. In movie-related graphs such as IMDB,

[1]Anonymous Institution, Anonymous City, Anonymous Region, Anonymous Country. Correspondence to: Anonymous Author <anon.email@domain.com>.

Preliminary work. Under review by the International Conference on Machine Learning (ICML). Do not distribute.

new entities and relations emerge over time, reflecting shifts in genres, collaborations, and audience interests. In such evolving environments, HGNNs are expected to continuously incorporate the incoming new knowledge while preserving the learnt patterns. However, existing continual learning (CL) methods, largely developed for images (Aljundi et al., 2018; Kirkpatrick et al., 2017; Li & Hoiem, 2017; Lopez-Paz & Ranzato, 2017) and homogeneous graphs (ZHANG et al., 2022; Liu et al., 2021; Zhou & Cao, 2021; Zhang et al., 2023a; Unal et al., 2023; Su et al., 2023; Ren et al., 2023; Niu et al., 2024; Li et al., 2024; Mondal et al., 2024; Liu et al., 2024; Choi et al., 2024; Qiao et al., 2025), fall short when directly applied to heterogeneous graphs due to semantic diversity and structural imbalance. For instance, in recommendation platforms, rare product categories are easily forgotten; in knowledge graphs, semantically distinct relations collapse into uniform representations; and in social networks, global regularization fails to capture type-specific parameter importance. These challenges suggest the necessity to develop new continual learning strategies which can explicitly account for real-world heterogeneity.

To bridge this critical gap, we present a systematic investigation of the Heterogeneous Continual Graph Learning (HCGL) problem and introduce HERO, a unified framework for continual learning on heterogeneous graphs. Unlike homogeneous graphs, the diverse node and edge types in heterogeneous graph introduce extra challenges to knowledge transfer across different tasks. To support fast and stable adaptation, we design a Gradient-based Meta-learning Module (G-MM). G-MM learns a shared initialization using gradient-based meta-learning. This initialization enables fast adaptation with few samples. Task-specific updates reduce interference across tasks. This improves robustness on past tasks. Memory replay has proven to be an effective strategy for mitigating catastrophic forgetting. However, existing replay methods typically rely on storing large volumes of historical data, which makes continual learning impractical in large-scale tasks. Moreover, conventional memory sampling strategies cannot be directly applied to HCGL due to the complexity of heterogeneous structures. To address this, we propose DiSCo (Diversity Sampling with semantic Consistency). DiSCo selects diverse target nodes and expands compact subgraphs along meta-paths. This strategy preserves key semantic and structural information with low memory cost. We also introduce a Heterogeneity-aware Knowledge Distillation (HKD) module. Replay alone is not sufficient to preserve knowledge. HKD aligns models across tasks at multiple levels. It matches prediction logits and semantic representations. Meta-path-based attention is also distilled. This alignment helps the model retain useful information while learning new tasks. Our contributions are summarized as follows:

1. We present a systematic study of Heterogeneous Continual Graph Learning and analyze its challenges beyond homogeneous graphs.

2. We propose DiSCo, a diversity- and semantics-aware sampling method for memory-efficient replay on heterogeneous graphs.

3. We design a heterogeneity-aware knowledge distillation approach that aligns information at logit and semantic levels.

4. We develop HERO, a unified framework for HCGL. Experiments on multiple heterogeneous graph benchmarks show strong gains in accuracy, efficiency, and robustness.

## 2. Related Work

### 2.1. Heterogeneous Graph Neural Networks

Heterogeneous Graph Neural Networks (HGNNs) are designed to model graphs with multiple node and edge types by encoding type-dependent structure and semantics. As discussed in the introduction, existing HGNNs mainly fall into metapath-based approaches (Fu et al., 2020; Zhang et al., 2019; Wang et al., 2019; Schlichtkrull et al., 2018; Dong et al., 2017; Sun et al., 2011; Yun et al., 2019; Yang et al., 2023; Zhu et al., 2024) and metapath-free approaches (Hu et al., 2020; Lv et al., 2021; Zhu et al., 2019; Hong et al., 2020; Zhang et al., 2022b; Fu et al., 2023; Huo et al., 2025). These methods have demonstrated strong performance in extracting semantic and structural information from heterogeneous graphs under static settings. While HGNNs have achieved remarkable success in learning from static heterogeneous graphs, they typically rely on the assumption that the entire graph is accessible during training. This assumption limits their applicability in dynamic real-world scenarios, where graph structures evolve continuously over time, as seen in knowledge graphs and social networks. Consequently, Continual Learning (CL) emerges as a critical yet underexplored research direction for advancing HGNNs in such dynamic settings.

### 2.2. Continual Graph Learning

Continual Graph Learning (CGL) studies how graph models can learn from sequentially arriving data while reducing catastrophic forgetting. Most existing CGL methods are developed under homogeneous graph settings, where all nodes and edges share the same type. As a result, their design assumptions and learning objectives differ substantially from those required in HCGL, which involves multiple node and relation types with distinct semantics. A key gap between CGL and HCGL lies in the nature of forgetting. In homogeneous graphs, forgetting mainly arises from distribution shift in node features or local topology. In contrast, HCGL

introduces additional sources of forgetting caused by semantic imbalance across types, uneven arrival of node or relation categories, and cross-type dependency shifts. Existing CGL methods do not explicitly model these heterogeneous semantics, making them less effective when previously learned type-specific knowledge is overwritten by newly introduced types or relations.

Parameter-isolation and regularization-based CGL methods (Zhang et al., 2022a; 2023a; Kirkpatrick et al., 2017; Aljundi et al., 2018; Liu et al., 2021) typically measure parameter importance at a global level. In heterogeneous graphs, however, parameters often serve different roles for different node or edge types. Treating all parameters uniformly can either over-constrain learning for new types or fail to protect critical parameters tied to rare but important semantics. Similarly, distillation-based approaches (Xu et al., 2020; Dong et al., 2021) usually align predictions or embeddings without accounting for type-specific feature spaces, which can lead to semantic drift in heterogeneous settings. Memory-replay methods (Zhang et al., 2023b; Ahrabian et al., 2021; Zhou & Cao, 2021; Zhang et al., 2022b; Choi et al., 2024; Zhang et al., 2024a; Hoang et al., 2023; Han et al., 2024; Song et al., 2025) face additional challenges in HCGL. Storing representative nodes or subgraphs becomes more complex when multiple types must be balanced within limited memory. Naive replay strategies may bias the buffer toward dominant types, while replaying full heterogeneous subgraphs increases both storage and computation cost. Moreover, replay mechanisms designed for homogeneous structures do not ensure semantic consistency across heterogeneous relations, which limits their ability to preserve cross-type knowledge over time.

These limitations indicate that HCGL requires mechanisms beyond direct extensions of existing CGL methods. In this work, we address this gap by introducing DiSCo and heterogeneity-aware knowledge distillation, which explicitly consider type-level semantics and cross-type interactions during continual learning on heterogeneous graphs.

### 2.3. Meta Learning

Meta-learning aims to enable models to adapt rapidly to new tasks by leveraging shared knowledge. Existing approaches are generally categorized into optimization-based (e.g., MAML (Finn et al., 2017), Reptile (Nichol & Schulman, 2018)) and metric-based methods (e.g., Prototypical Networks (Snell et al., 2017), Matching Networks (Vinyals et al., 2016)). In continual learning, meta-learning has been applied to online (Gupta et al., 2020) and few-shot scenarios (Javed & White, 2019). Recently, its integration with graph learning has drawn increasing attention. For example, MetaCLGraph (Unal et al., 2023) incorporates meta-learning with experience replay for continual graph learning,

and HG-Meta (Zhang et al., 2022a) applies it to few-shot graph classification on heterogeneous graphs. Motivated by these advances, we further explore its potential in mitigating forgetting and improving adaptability in heterogeneous continual learning.

### 3. Preliminary

In this section, we introduce fundamental concepts and formalize the problem of HCGL. Our primary objective is to develop a continual learning framework on Evolving Heterogeneous Graphs to mitigate the issue of catastrophic forgetting. In the HCGL setting, a HGNN learns a sequence of tasks in a domain incremental setting (Zhang et al., 2024b) (See Appendix B.3 for the details of this setting), without access to the data from previous tasks. However, it is allowed to utilize a memory buffer with limited capacity to store representative information. The goal of the framework is to maximize the prediction accuracy across all tasks after training while minimizing the forgetting of previously acquired knowledge during the learning of new tasks.

In this work, we focus on node classification tasks and adopt a common continual learning setup in which the dataset is partitioned into a sequence of tasks based on node class labels. In this way, different splits have non-overlapping category labels.

**Definition 3.1.** A heterogeneous graph is defined as $G = (V, E)$, where $V$ is the set of nodes, and $E$ is the set of edges. Each node $v \in V$ is associated with a type given by the node-type mapping function $\phi : V \to \mathcal{A}$, and each edge $e \in E$ is associated with a type given by the edge-type mapping function $\psi : E \to \mathcal{R}$, where $\mathcal{A}$ and $\mathcal{R}$ denote the sets of node and edge types, respectively. The graph is considered heterogeneous if $|\mathcal{A}| + |\mathcal{R}| > 2$. Let $V_\tau$ denote the set of nodes of type $\tau \in \mathcal{T}$.

**Definition 3.2.** A metapath $\mathcal{P}$ is a sequence of node and edge types in the form of $A_1 \xrightarrow{R_1} A_2 \xrightarrow{R_2} \cdots \xrightarrow{R_l} A_{l+1}$, which defines a composite relation $R_1 \circ R_2 \circ \cdots \circ R_l$ between node types $A_1$ and $A_{l+1}$, where $\circ$ denotes the composition operator. The metapath captures high-level semantic connections across different types of nodes through a specific sequence of edge types.

**Definition 3.3.** Let $\mathcal{G} = (V, E)$ be a dynamic heterogeneous graph, where $V$ is the node set and $E$ is the edge set. The feature set is type-specific, i.e., $\mathcal{F} = \{F_\tau \in \mathbb{R}^{|V_\tau| \times d_\tau} \mid \tau \in \mathcal{A}\}$. In the HCGL setting, the model learns a sequence of tasks $\{\mathcal{T}_1, \mathcal{T}_2, \ldots, \mathcal{T}_T\}$, without access to the data from previous tasks. Specifically, each task $\mathcal{T}_t$ corresponds to a subgraph $\mathcal{G}_t$ with disjoint category labels, and due to storage limitations, the model can only access the data available at the current task. For each subgraph $\mathcal{G}_t$, we divide it into a training set $\mathcal{G}_t^{\text{tr}} = (V_t^{\text{tr}}, E_t^{\text{tr}})$ and a testing set $\mathcal{G}t^{\text{te}} = (V_t^{\text{te}}, E_t^{\text{te}})$ to train and evaluate the model $f_\theta$.

# 4. Methodology

In this section, we present our HERO (HEterogeneous continual gRaph learning via meta-knOwledge distillation) framework (see Figure 2), provide a detailed introduction of its core components, and explain how it can systematically address the identified challenges.

## 4.1. Fast Adaptation with Gradient-based Meta-Learning

Efficient adaptation to new data patterns is crucial for ensuring the practical usability of a model in an evolving heterogeneous web graph. To achieve this, we introduce the Gradient-based Meta-learning Module (G-MM), which optimizes model parameter initialization to enable rapid adaptation to new tasks. This, in turn, helps preserve performance on the current task. Specifically, given the training node set $V_t^{tr}$ of the current task $\mathcal{T}_t$, we select $e$ samples from $V_t^{tr}$ based on the Coverage Maximization (CM) strategy (introduced in Section 4.2), which identifies key nodes by maximizing node diversity. In practice, labeled data in heterogeneous graphs is often highly limited. By performing gradient descent on the selected small sample set $\mathcal{E}$, the model can quickly adapt to the current task, a process referred to as the inner update. Let $\theta$ denote the set of model parameters. For the current task $\mathcal{T}_t$, We feed the sampled node set $\mathcal{E}$ into the model and compute the loss $\mathcal{L}_\mathcal{E}$, updating $\theta$ via gradient descent (Inner Update):

$$\theta \leftarrow \theta - \alpha \nabla_\theta \mathcal{L}_\mathcal{E}(\theta), \quad (1)$$

where $\alpha$ is the step size for the inner update. Meta-learning with a small number of samples enables the model to prevent overfitting during experience replay while ensuring rapid adaptation to the current task, thereby maintaining robust overall performance.

## 4.2. Diversity Sampling with semantic Consistency

In addition to adapting to new data patterns, overcoming catastrophic forgetting is crucial, as previously observed patterns may reappear in practical scenarios. While many existing experience replay methods primarily consider homogeneous graphs or single-type nodes, they fail to capture the complex semantic and structural dependencies in heterogeneous graphs. To address this, we propose DiSCo (Diversity Sampling with semantic Consistency), which jointly considers the diversity of nodes within the target node type and the structural connections with nodes of other types. DisCo proceeds in two stages: (1) selecting representative target-type nodes by maximizing diversity in the feature or embedding space, and (2) expanding to other types of nodes through relation-type-aware importance estimation, ensuring that both semantic diversity and heterogeneous structural patterns are preserved.

**Step 1: Target Node Selection via Coverage Maximization.** Prior research has shown that different data points contribute unequally to model training: some significantly enhance performance, while others offer limited benefit (Toneva et al., 2018). Motivated by this, we identify key nodes from the current task by maximizing node diversity using the Coverage Maximization (CM) strategy (Zhou & Cao, 2021). This approach allows us to retain essential historical information with only a small number of nodes. Formally, given the training node set $V_t^{tr}$ of task $\mathcal{T}_t$, CM selects a subset by maximizing the coverage of the attribute/embedding space:

$$\mathcal{N}(v_i) = \{v_j \mid \text{dist}(v_i, v_j) < d, \mathcal{Y}(v_i) \neq \mathcal{Y}(v_j)\}, \quad (2)$$

where $\mathcal{Y}(v_i)$ denotes the label of node $v_i$, and $\mathcal{N}(v_i)$ represents the set of nodes from different classes that are within a distance $d$ of $v_i$. Given the memory buffer $\mathcal{B}_{\tau_t}$ for the target node type $\tau_t$, we select $e$ nodes per class with the smallest $|\mathcal{N}(v_i)|$ as representative experiences.

**Step 2: Multi-type Neighbor Expansion.** Given target nodes $V_{\tau_t}$, we collect candidate nodes $\mathcal{C}_\tau$ for node type $\tau$ with relation type $r = (\tau, \tau_t)$ through:

$$\mathcal{C}_\tau = \bigcup_{v \in V_{\tau_t}} \{u \mid (v, u) \in E_r \vee (u, v) \in E_r\}, \quad (3)$$

where $E_r$ denotes edges of relation type $r$. We define node importance $\pi(v)$ as the sum of relation-specific degrees:

$$\pi(v) = \sum_{r \in \mathcal{R}} \deg_r(v), \quad (4)$$

where $\deg_r(v)$ measures the connectivity strength of node $v$ under relation type $r$. For each node type $\tau$ with buffer $\mathcal{B}_\tau$, we select top-$|\mathcal{B}_\tau|$ nodes by $\pi(v)$:

$$V_\tau^* = \text{topk}_{v \in \mathcal{C}_\tau}(\pi(v), |\mathcal{B}_\tau|), \quad (5)$$

With sampled target type node set $V_{\tau_t}$ and the selected heterogeneous neighbor nodes set $\{V_\tau\}_{\tau \in \mathcal{A}/\tau_t}$, we construct the heterogeneous subgraph $\mathcal{G}_{sub} = (V', E')$ through:

$$V' = V_{\tau_t} \cup \bigcup_{\tau \neq \tau_t} V_\tau, \quad E' = \{(u, v) \in E \mid u, v \in V'\},$$
$$(6)$$

For all $v \in V_{\tau_t}$, we retain their original features and labels, and maintain the structure of the type-specific projection. When learning a new task, we perform the experience replay by incorporating an auxiliary loss computed over the memory buffer $\mathcal{B}$ into the current task loss (i.e., the cross-entropy loss between the given labels $\mathcal{Y}$ and the predicted labels $\hat{\mathcal{Y}}$)

$$\mathcal{L} = \mathcal{L}_{\mathcal{T}_t}(\mathcal{G}_t, \theta) + \lambda_{er} \mathcal{L}_{er}(\mathcal{G}_{sub}, \theta) \quad (7)$$
$$= -\sum_{v_i \in \mathcal{V}_t^{tr}} \mathcal{Y}(v_i) \log \hat{\mathcal{Y}}(v_i) - \lambda_{er} \sum_{v_j \in \mathcal{B}} \mathcal{Y}(v_j) \log \hat{\mathcal{Y}}(v_j),$$

where $\hat{\mathcal{Y}}(v_i)$ is the predicted label of node $v_i$, $\lambda_{er}$ modulates the contribution of buffered data in the overall loss.

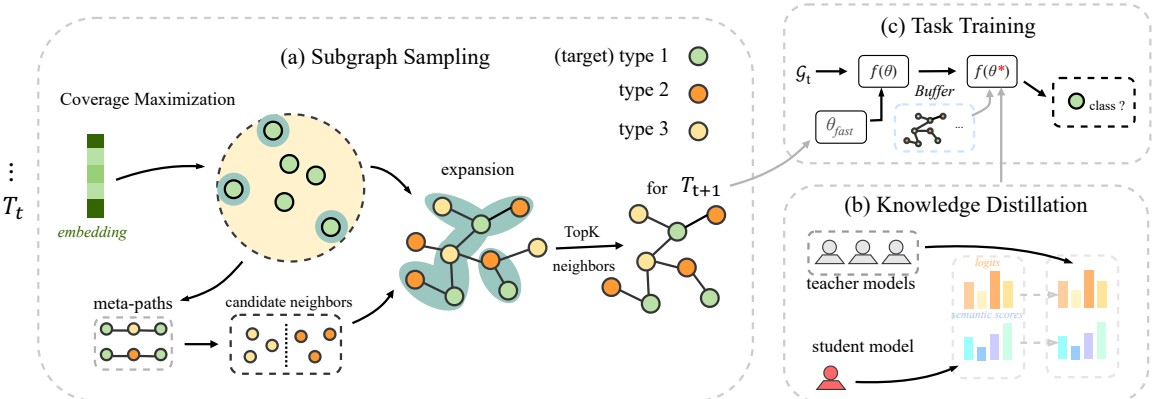

*Figure 2.* The overall framework of HERO. (a) Diversity Sampling with semantic Consistency: Construct task-specific subgraphs by selecting diverse target-type nodes and expanding to related node types via relation-aware importance. (b) Heterogeneity-aware Knowledge Distillation: Align previous and current tasks via logit-level and semantic-level distillation. (c) Task Training: Use meta-learning for fast adaptation, jointly optimizing task loss, replay loss, and distillation loss.

### 4.3. Heterogeneity-aware Knowledge Distillation for Task Alignment

Existing knowledge distillation methods are often developed for homogeneous scenarios and fail to account for the multi-relational semantics and cross-type dependencies in heterogeneous graphs. In contrast, our Heterogeneity-aware Knowledge Distillation (HKD) module is explicitly tailored for heterogeneous graph continual learning. Instead of relying solely on logit-based distillation, we propose a two-level alignment strategy that leverages both prediction and semantic signals to capture graph heterogeneity. The logit-level distillation transfers soft label distributions to retain discriminative knowledge, while the semantic-level alignment matches meta-path-aware attention scores, enabling the student model to preserve high-order structural patterns unique to heterogeneous graphs.

**Logit-level Distillation** Inspired by previous distillation-based approaches (Tian et al., 2023), we transfer the soft knowledge from the teacher model (previous task classifier) to the student model (current task classifier), enabling the student model to learn both new knowledge from the current task and retained knowledge from past tasks. Mimicking teacher's prediction results enables the student model to learn the secondary information from previous tasks that cannot be expressed by the experience replay data alone. Soft knowledge from teacher model is formulated as the predicted probability of the labels in the current task data:

$$P^T(z_i, t) = \text{Softmax}\big(f_T(z_i), t\big) = \frac{\exp\left[f_T(z_i)/t\right]}{\sum_j \exp\left[f_T(z_j)/t\right]}, \quad (8)$$

where $z_i$ is the embedding of node $v_i$ in $\mathcal{V}_i^{tr}$, $f_T(z_i)$ is the score logit of $z_i$ obtained from teacher model, and $t$ is the

temperature index to soften the peaky softmax distribution (Hinton, 2015). Thus, the knowledge distillation loss of the teacher model and the student model is defined as follows:

$$\mathcal{L}_{\text{logit}} = \frac{t^2}{N} \sum_l^N \sum_{v_i \in \mathcal{V}_i^{tr}} P_l^T(z_i, t) \log \frac{P_l^T(z_i, t)}{P^S(z_i, t)}, \quad (9)$$

where $N$ is the number of teacher models, $P^T$ and $P^S$ are the predicted distributions of teacher model and student model respectively.

**Semantic-level Distillation** To preserve metapath-induced semantic patterns, we propose an attention-based structural distillation method. Given a predefined metapath set $\mathcal{P} = \{P_1, \dots, P_M\}$, let $\boldsymbol{\alpha}_{P_m}^T(v_i)$ and $\boldsymbol{\alpha}_{P_m}^S(v_i)$ denote the attention coefficients for metapath $P_m$ computed by the teacher and student models, respectively. These attention coefficients reflect the importance of different semantic contexts encoded by heterogeneous structural patterns. To align the semantic-level knowledge between the teacher and the student, we define the semantic alignment loss as:

$$\mathcal{L}_{\text{sem}} = \frac{1}{N} \sum_{l=1}^N \sum_{m=1}^M \|\boldsymbol{\alpha}_{l, P_m}^T - \boldsymbol{\alpha}_{P_m}^S\|_2 \quad (10)$$

where $\boldsymbol{\alpha}_{P_m} \in \mathbb{R}^{|V^{tr}|}$ is the normalized attention vector over all nodes for metapath $P_m$. We provide a detailed discussion in Appendix A.1 on how this module can be extended to arbitrary HGNNs beyond metapath-based architectures.

This loss guides the student model to capture relational importance and structural dependencies aligned with the teacher model, thereby preserving semantic information that is critical in heterogeneous graphs. Afterwards, we

combine the logit level loss and the semantic level loss:

$$\mathcal{L}_{kd} = \lambda_{\text{logit}}\mathcal{L}_{\text{logit}} + \lambda_{\text{sem}}\mathcal{L}_{\text{sem}}, \tag{11}$$

where $\lambda_{\text{logit}}$ and $\lambda_{\text{sem}}$ are both trade-off weights for balancing the losses. The final objective of node representation learning is to minimize the joint loss including current task loss $\mathcal{L}_{\mathcal{T}_i}$, experience replay loss $\mathcal{L}_{er}$, and the KD loss $\mathcal{L}_{kd}$:

$$\mathcal{L}_{joint} = \mathcal{L}_{\mathcal{T}_t} + \lambda_{er}\mathcal{L}_{er} + \lambda_{kd}\mathcal{L}_{kd}, \tag{12}$$

where $\lambda_{kd}$ is a trade-off weight for balancing the losses. Taking the parameters $\theta$ obtained from Equation 1 as initialization, the student model for the current task is further updated as follows:

$$\theta^* = \theta - \beta\nabla_\theta\mathcal{L}_{joint}(\theta), \tag{13}$$

where $\beta$ is learning rate. By minimizing $\mathcal{L}_{joint}$, parameters $\theta$ of HGNN model are optimized for downstream node classification task. As the number of tasks grows, the teacher models would increase linearly. For scalability, we adapt a sliding window strategy that only keeps the latest three teacher models for knowledge distillation.

## 5. Experiments

We conducted experiments on four widely used benchmarks: DBLP (Lv et al., 2021), IMDB (Lv et al., 2021), Freebase (Lv et al., 2021) and Yelp (Yang et al., 2020) to evaluate the performance of HERO. We adopt three widely used HGNN backbones in our experiments: HAN (Wang et al., 2019), MAGNN (Fu et al., 2020), and HGT (Hu et al., 2020). We compared our method with state-of-the-art baselines. Due to setting mismatch, existing continual graph learning methods cannot be directly applied, especially memory-replay approaches. Following prior work, we sample target-type nodes using their original strategies, while sampling other node types with our method to ensure compatibility and fair comparison. See Appendix B for the details of the datasets, baselines and experiment setup.

### 5.1. Metrics

In our experiments, *Average Performance* (AP) and *Average Forgetting* (AF) (Lopez-Paz & Ranzato, 2017), are used to measure the performance on test sets. AP and AF are defined as $\text{AP} = \frac{1}{T}\sum_{t=1}^{T}a_{T,t}$, $\text{AF} = \frac{1}{T-1}\sum_{t=1}^{T-1}(a_{t,t} - a_{T,t})$, where $T$ is the total number of tasks and $a_{i,j}$ is the accuracy of the model on the test set of task $j$ after it is trained on task $i$.

### 5.2. Performance Evaluation

The overall performance on three heterogeneous graph benchmarks is reported in Table 1. Task-wise accuracy

retention results on Yelp are further illustrated in Figure 3 and Figure 6. In addition, we conduct extended experiments to assess the robustness of HERO, with detailed results provided in Appendix D. We highlight the following key observations: **(1)** All HGNNs exhibit catastrophic forgetting on previous tasks. For example, on the DBLP dataset, the average forgetting (AF) for HAN and MAGNN is over 26% and 34%, respectively; on the IMDB dataset, the AF for all HGNNs is over 26%, 17%, and 21%, respectively. **(2)** Across all three HGNN backbones, HERO substantially outperforms state-of-the-art CGL methods in terms of average performance (AP) while maintaining the lowest forgetting (AF). For example, on DBLP with HGT backbone, HERO improves AP by +2.1% over the strongest baseline (ERGNN), close to the upper bound of joint training. Similar consistent performance gains are observed on IMDB and Freebase. These results confirm the effectiveness of HERO as a holistic framework for balancing knowledge adaptation and retention in HCGL. **(3)** It is noteworthy that, in some cases, memory-replay based methods (e.g., FTF-ER, DMSG, and ER-GNN) perform better than HERO in terms of AF, despite their lower AP scores compared to HERO. This is because these methods rely on retraining nodes from previously learned tasks. Once enough nodes are sampled, these models can largely mitigate catastrophic forgetting. However, this also sacrifices performance when learning new tasks. Although these methods attempt to address the new-old task trade-off through different strategies (e.g., ER-GNN reweights the loss between new and old tasks), they still cannot fully avoid the performance drop. We further discuss this in Appendix G. **(4)** We also observe a notable phenomenon: in the 3-way setting, memory-replay methods generally beat regularization-based methods, while in the 2-way setting the opposite holds. Further analysis of Figure 6 shows that regularization methods are less effective than memory-replay at overcoming forgetting of earlier tasks, but they achieve better performance on recent tasks, which is the "performance sacrifice" effect mentioned earlier. Memory-replay tends to accumulate redundant or conflicting information over long task sequences, which can harm adaptation to recent tasks. HERO mitigates this by filtering redundant information and applying knowledge distillation, preserving historical knowledge while maintaining stronger performance on recent tasks.

### 5.3. Ablation Study

We evaluate the contribution of each HERO component using HAN as the backbone across four benchmark datasets (Table 2). We first examined the impact of each component by sequentially removing Experience Replay (ER), Heterogeneity-aware Knowledge Distillation (HKD), and the Gradient-based Meta-learning Module (G-MM) while keeping other components intact. The results demonstrate

*Table 1.* Performance comparison with different HGNN backbones on three benchmark datasets. The symbol ↑ (↓) means higher (lower) is better. The best results are highlighted in **bold**, while the second best results are underlined. "OOM" means that the model runs out of memory on large graphs.

| Base Models | Methods | DBLP | | IMDB | | Freebase | |
|---|---|---|---|---|---|---|---|
| | | AP / % ↑ | AF / % ↓ | AP / % ↑ | AF / % ↓ | AP / % ↑ | AF / % ↓ |
| HAN | Finetune | 82.8 ± 4.6 | 26.6 ± 9.1 | 68.8 ± 2.2 | 26.0 ± 2.8 | 53.8 ± 4.7 | 25.7 ± 8.8 |
| | Joint Train | 95.2 ± 0.6 | 1.7 ± 1.0 | 80.4 ± 0.7 | 3.9 ± 0.8 | 65.6 ± 0.9 | 2.8 ± 2.0 |
| | EWC (Kirkpatrick et al., 2017) | 85.4 ± 5.5 | 21.5 ± 10.6 | 69.6 ± 1.2 | 25.8 ± 2.4 | 58.3 ± 2.4 | 18.4 ± 4.2 |
| | MAS (Aljundi et al., 2018) | 91.1 ± 0.8 | 8.3 ± 1.5 | 72.0 ± 2.3 | 20.5 ± 3.9 | 59.0 ± 1.0 | 14.8 ± 0.9 |
| | TWP (Liu et al., 2021) | 90.5 ± 1.3 | 10.1 ± 2.6 | 74.8 ± 2.7 | 14.5 ± 4.4 | 60.9 ± 4.3 | **10.8 ± 6.9** |
| | ER-GNN (Zhou & Cao, 2021) | 89.7 ± 2.9 | 13.1 ± 6.1 | 74.9 ± 2.3 | 12.7 ± 4.2 | 56.5 ± 1.3 | 19.6 ± 0.9 |
| | MetaCLGraph (Unal et al., 2023) | 89.9 ± 0.9 | 11.4 ± 2.3 | 74.8 ± 3.2 | 14.8 ± 5.7 | 59.4 ± 2.5 | 13.0 ± 4.1 |
| | FTF-ER (Pang et al., 2024) | 90.8 ± 1.4 | 9.8 ± 2.4 | 72.0 ± 4.2 | 19.7 ± 6.4 | 58.1 ± 0.7 | 14.6 ± 0.6 |
| | DMSG (Qiao et al., 2025) | 93.5 ± 0.6 | **0.7 ± 0.6** | 68.8 ± 3.9 | 25.1 ± 7.8 | 57.1 ± 2.4 | 17.5 ± 3.6 |
| | Ours | **93.6 ± 0.8** | 4.3 ± 1.6 | **78.4 ± 1.7** | **7.3 ± 3.0** | **60.9 ± 1.5** | 11.1 ± 1.6 |
| HGT | Finetune | 86.2 ± 1.5 | 12.7 ± 2.2 | 75.1 ± 7.5 | 17.0 ± 9.4 | 67.8 ± 1.8 | 16.5 ± 2.5 |
| | Joint Train | 93.9 ± 0.6 | 1.7 ± 1.0 | 80.0 ± 0.7 | 2.0 ± 0.9 | 72.5 ± 0.6 | 8.5 ± 1.2 |
| | EWC (Kirkpatrick et al., 2017) | 89.1 ± 1.2 | 10.7 ± 1.8 | 77.3 ± 0.4 | 12.5 ± 0.2 | 69.6 ± 3.0 | 14.7 ± 2.9 |
| | MAS (Aljundi et al., 2018) | 89.8 ± 2.5 | 9.0 ± 3.0 | 77.5 ± 1.2 | 11.8 ± 2.6 | 70.4 ± 3.1 | 14.8 ± 2.0 |
| | TWP (Liu et al., 2021) | 89.1 ± 2.1 | 10.3 ± 1.1 | 77.7 ± 0.6 | 11.1 ± 1.6 | 69.6 ± 2.0 | 15.1 ± 0.9 |
| | ER-GNN (Zhou & Cao, 2021) | 90.7 ± 1.5 | 3.5 ± 0.4 | 77.3 ± 2.2 | 9.0 ± 3.6 | 69.5 ± 1.4 | 13.6 ± 1.7 |
| | MetaCLGraph (Unal et al., 2023) | 90.0 ± 0.7 | 10.9 ± 1.4 | 76.1 ± 0.3 | 12.2 ± 1.8 | 69.1 ± 2.2 | 16.3 ± 2.5 |
| | FTF-ER (Pang et al., 2024) | 89.4 ± 2.2 | 5.5 ± 1.9 | 76.1 ± 1.4 | 8.9 ± 3.5 | 69.7 ± 2.8 | 13.0 ± 2.5 |
| | DMSG (Qiao et al., 2025) | 88.2 ± 1.9 | 11.3 ± 2.4 | 76.1 ± 0.4 | 12.1 ± 1.5 | 64.6 ± 0.6 | 16.4 ± 2.7 |
| | Ours | **92.8 ± 0.8** | **3.3 ± 1.2** | **78.4 ± 0.4** | **6.6 ± 1.7** | **70.6 ± 1.2** | **9.1 ± 2.7** |
| MAGNN | Finetune | 79.1 ± 5.1 | 34.8 ± 10.1 | 71.3 ± 0.6 | 21.7 ± 3.0 | OOM | OOM |
| | Joint Train | 94.1 ± 0.1 | 1.9 ± 1.0 | 77.4 ± 0.7 | 4.7 ± 1.5 | OOM | OOM |
| | EWC (Kirkpatrick et al., 2017) | 91.0 ± 2.3 | 10.9 ± 4.5 | 73.4 ± 1.6 | 16.7 ± 2.5 | OOM | OOM |
| | MAS (Aljundi et al., 2018) | 92.1 ± 2.1 | 7.8 ± 5.3 | 74.3 ± 2.0 | 14.2 ± 3.7 | OOM | OOM |
| | TWP (Liu et al., 2021) | 91.6 ± 2.1 | 7.4 ± 5.0 | 74.0 ± 0.8 | 13.7 ± 3.3 | OOM | OOM |
| | ER-GNN (Zhou & Cao, 2021) | 90.0 ± 2.1 | 12.4 ± 4.3 | 75.3 ± 0.9 | 10.3 ± 1.5 | OOM | OOM |
| | MetaCLGraph (Unal et al., 2023) | 92.3 ± 0.4 | 2.1 ± 0.7 | 75.6 ± 1.0 | 11.8 ± 1.3 | OOM | OOM |
| | FTF-ER (Pang et al., 2024) | 92.8 ± 0.8 | **1.9 ± 2.3** | 73.0 ± 0.6 | 15.8 ± 1.4 | OOM | OOM |
| | DMSG (Qiao et al., 2025) | 90.6 ± 1.1 | 8.6 ± 2.2 | 74.5 ± 0.5 | 8.8 ± 2.7 | OOM | OOM |
| | Ours | **93.2 ± 0.5** | 3.1 ± 0.8 | **76.8 ± 1.1** | **7.5 ± 1.4** | OOM | OOM |

that removing any module leads to performance drops in both AP and AF, confirming each module's importance. Notably, the removal of G-MM shows relatively smaller performance impact (0.3-3.4% decrease in AP compared to 1.5-11.0% for ER/HKD), which aligns with its primary function of facilitating rapid adaptation to new tasks rather than long-term knowledge retention, the latter being mainly handled by the synergistic effect of ER and HKD. Particularly, the absence of HKD causes the most significant performance deterioration (3.6-12.0% increase in FR), highlighting its crucial role as the core mechanism against catastrophic forgetting. Furthermore, we replaced our proposed DiSCo with the basic Coverage Maximization (CM) strategy. As shown in the last row of Table 2, DiSCo demonstrates substantial performance advantages, especially on edge-dense datasets like Yelp. These results confirm that DiSCo significantly outperforms conventional sampling methods in preserving the original topological structure of heterogeneous graphs. Detailed analyses of HKD submodules are provided in Appendix E.

## 5.4. Hyperparameters Analysis

We analyze the impact of key hyperparameters on model performance, including the loss weight ($\lambda_{kd}$) in knowledge distillation (KD), the sampling budget for experience replay, and the shot number (i.e., the number of sampled nodes used in meta-learning), as shown in Figure 4. We further discuss in Appendix F the impact of the trade-off weight used to balance the losses of two submodules in the knowledge distillation component. The experimental findings are as follows: **Distillation loss Weight** ($\lambda_{kd}$): On IMDB, higher weights better support knowledge transfer, while lower values ($\lambda_{kd} < 0.2$) lead to increased forgetting. On Freebase, smaller weights ($0.1 \sim 0.3$) help mitigate forgetting, but larger weights ($\lambda_{kd} > 0.3$) hinder adaptability. For Yelp, the 3-way setting introduces large task shifts, requiring higher weights for stability and effective forgetting mitigation. **Sample Budget**: The sample budget influences the stability-plasticity trade-off. Too few nodes lead to insufficient information, degrading performance, while excessive sampling limits the model's adaptability

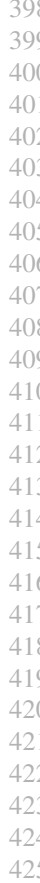
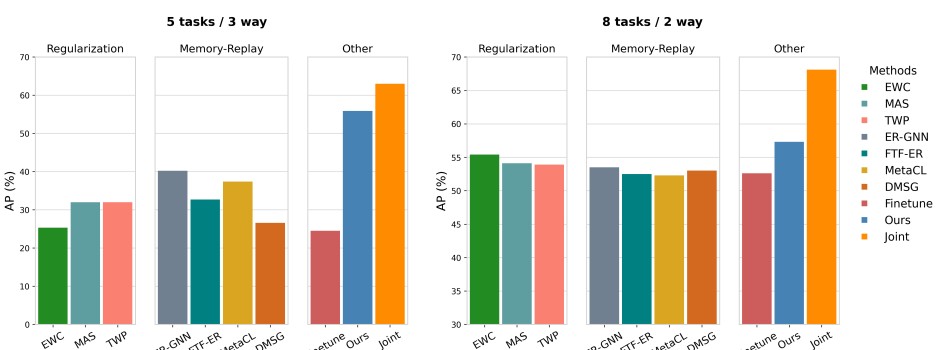

*Figure 3.* Performance comparison on the Yelp dataset under two settings using HAN as the backbone model.

*Table 2.* Ablation results of HERO using HAN as the backbone. We remove Experience Replay (ER), Heterogeneity-aware Knowledge Distillation (HKD), and the Gradient-based Meta-learning Module (G-MM) individually. And "w/o DiSCo" denotes replacing the proposed *Diversity Sampling with semantic Consistency* with Coverage Maximization applied to all node types.

| Methods | DBLP | | IMDB | | Freebase | | Yelp | |
| --- | --- | --- | --- | --- | --- | --- | --- | --- |
| | AP / % ↑ | AF / % ↓ | AP / % ↑ | AF / % ↓ | AP / % ↑ | AF / % ↓ | AP / % ↑ | AF / % ↓ |
| HERO | **93.6 ± 0.8** | **4.3 ± 1.6** | **78.4 ± 1.7** | **7.3 ± 3.0** | **60.9 ± 1.5** | **11.1 ± 1.6** | **55.9 ± 0.6** | **7.5 ± 1.2** |
| w/o ER | 89.7 ± 0.6 | 12.0 ± 0.7 | 75.1 ± 0.9 | 12.8 ± 1.7 | 58.6 ± 1.2 | 16.3 ± 2.2 | 53.9 ± 2.4 | 6.8 ± 2.4 |
| w/o HKD | 89.6 ± 1.4 | 12.1 ± 2.6 | 72.2 ± 1.5 | 19.3 ± 2.3 | 59.4 ± 1.1 | 14.7 ± 1.8 | 44.9 ± 14.3 | 19.3 ± 17.0 |
| w/o G-MM | 92.6 ± 1.7 | 5.2 ± 0.9 | 75.0 ± 2.7 | 13.3 ± 4.5 | 59.7 ± 0.9 | 12.0 ± 2.2 | 55.6 ± 1.9 | 8.0 ± 2.0 |
| w/o DiSCo | 92.4 ± 1.4 | 6.0 ± 3.1 | 75.4 ± 2.2 | 12.1 ± 5.0 | 56.2 ± 0.4 | 18.0 ± 0.5 | 45.7 ± 14.0 | 23.7 ± 15.6 |

to new tasks. As shown in Figure 4, on datasets like Yelp, a large sampling budget (e.g., > 20) significantly impairs learning on new tasks due to the accumulation of historical information. **Shot Number**: The number of sampled nodes impacts the stability-plasticity trade-off. Meta-learning enables rapid adaptation to new tasks. As shown in the results, model performance steadily improves up to 10-shot, but declines at 20-shot, likely due to overfitting to the current task's data patterns. The optimal shot number is 10, balancing historical information retention and computational efficiency.

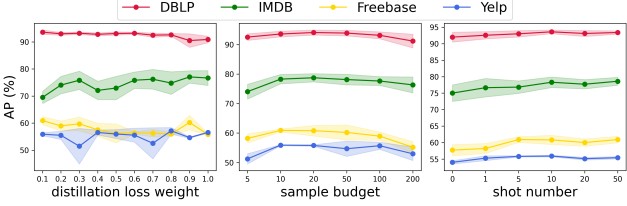

*Figure 4.* Sensitivity analysis of HERO with respect to key hyper-parameters. The shaded areas represent variances. From left to right: (1) distillation loss weight, (2) experience replay sampling budget, and (3) shot number used in meta-learning. Due to space constraints, we only report the Average Performance (AP) here.

### 5.5. Scalability Analysis

Based on the HAN backbone, we evaluate recent continual graph learning methods and the proposed HERO framework on four datasets using a single RTX 3080Ti GPU. Table 4 reports the average training time per task. Overall, regularization-based methods such as TWP are more efficient than replay-based approaches, highlighting a known ef-

ficiency gap in replay-driven continual learning. Addressing this gap in heterogeneous settings remains an open problem. Despite relying on replay, HERO is consistently faster than other replay-based methods, showing clear gains in training efficiency. Although TWP is slightly more efficient in terms of runtime, it underperforms HERO across all benchmarks with respect to accuracy and forgetting. These results indicate that HERO achieves a favorable balance between computational cost and learning performance, making it suitable for large-scale and long-sequence HCGL scenarios. Due to space constraints, we defer a detailed scalability analysis and additional results to Appendix C.

## 6. Conclusion

In this paper, we study heterogeneous continual graph learning (HCGL), a learning setting where both graph structure and node semantics evolve over time and heterogeneity is intrinsic. We propose HERO, a continual learning framework that explicitly accounts for heterogeneity in dynamic graphs. HERO consists of two core components. DiSCo is a replay strategy that selects diverse nodes and expands replay subgraphs along metapaths, which helps preserve semantic and structural information under strict memory constraints. This allows HERO to maintain competitive performance with substantially smaller replay buffers than prior methods. In addition, heterogeneity-aware distillation aligns representations across learning stages, reducing semantic drift and improving knowledge retention over time. Empirical results on multiple benchmarks show that these components work together to improve stability and adaptability in HCGL.

## Impact Statement

Heterogeneous graphs are widely used to model complex relational systems in critical domains such as biomedical networks, social platforms, and recommender systems. Our work on Heterogeneous Continual Graph Learning (HCGL) contributes to improving the robustness and adaptability of graph models under dynamic environments. This can benefit downstream applications such as real-time recommendation, dynamic knowledge graph completion, and biomedical discovery, where information constantly evolves. By enabling efficient knowledge retention and adaptation across tasks, our framework promotes safer and more reliable deployment of graph-based models in real-world systems.

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

# A. Methodology

## A.1. Extension to General HGNNs

Previously, we have discussed how to align semantic-level information via knowledge distillation in meta-path-guided attention networks. In this section, we further demonstrate that the proposed method can be easily extended to HGNNs without attention mechanisms (such as RGCN), enabling the preservation of semantic information from previous tasks.

For each pair of target-type nodes $v_i, v_j \in V_{\tau_t}$, we define the attention score $e_{ij}$ based on their hidden representations from the penultimate layer of the network $h_i^{(L-1)}$ and $h_j^{(L-1)}$, using the weight matrix $W^{(L)}$ from the last layer:

$$e_{ij} = \left( h_i^{(L-1)} W^{(L)} \right)^\top \tanh \left( h_j^{(L-1)} W^{(L)} \right), \tag{14}$$

We then apply softmax normalization to the attention scores for each node $v_i$ to obtain an attention vector:

$$\boldsymbol{\alpha}_i = \text{softmax}(e_{i1}, e_{i2}, \ldots, e_{i|V_{\tau_t}|}) \tag{15}$$

We compute attention vectors for both the teacher model $\boldsymbol{\alpha}_i^T$ and the student model $\boldsymbol{\alpha}_i^S$, and define the semantic alignment loss as the $L_2$ distance between the two:

$$\mathcal{L}_{\text{sem}} = \sum_{i \in V_{\tau_t}} \left\| \boldsymbol{\alpha}_i^T - \boldsymbol{\alpha}_i^S \right\|_2 \tag{16}$$

In addition to the node similarity-based attention defined above, for edge-type-aware attention models such as HGT, attention is computed for each relation type $r \in \mathcal{R}$. Let $\boldsymbol{\beta}_r^{(T)}$ and $\boldsymbol{\beta}_r^{(S)}$ denote the attention vectors for relation $r$ from the teacher and student models, respectively. The semantic alignment loss for such edge-type-based attention mechanisms is formulated as:

$$\mathcal{L}_{\text{sem}} = \sum_{r \in \mathcal{R}} \left\| \boldsymbol{\beta}_r^T - \boldsymbol{\beta}_r^S \right\|_2 \tag{17}$$

This formulation ensures that the student model preserves relation-aware structural semantics by aligning its attention distribution with that of the teacher model on each edge type.

*Table 3.* Details of datasets and continual learning tasks setting

| Node Classification | #Nodes | #Node Types | #Edges | #Edge Types | #Classes |
|---|---|---|---|---|---|
| DBLP | 26,128 | 4 | 239,566 | 6 | 4 |
| IMDB | 21,420 | 4 | 86,642 | 6 | 5 |
| Freebase | 180,098 | 8 | 1,057,688 | 36 | 7 |
| Yelp | 82,465 | 4 | 32,548,358 | 7 | 16 |

# B. Supplemental experiment setups

## B.1. Details of dataset

DBLP is a bibliographic dataset in the field of computer science. A widely adopted subset is used, containing four research areas, where nodes represent authors, papers, terms, and venues.

IMDB is a dataset derived from a online movie information platform. A subset including classes such as Action, Comedy, Drama, Romance, and Thriller is selected for use.

Freebase is a large-scale knowledge graph originally constructed for web search and semantic applications. A subgraph comprising eight genres of entities and approximately one million edges is sampled following procedures outlined in prior studies.

Yelp is a review network constructed from businesses, users, locations, and reviews. Although node features are not available, many business nodes are annotated with one or more labels from sixteen predefined categories. As a prototypical web application graph, Yelp embodies real-world challenges in recommendation and user–business interaction modeling.

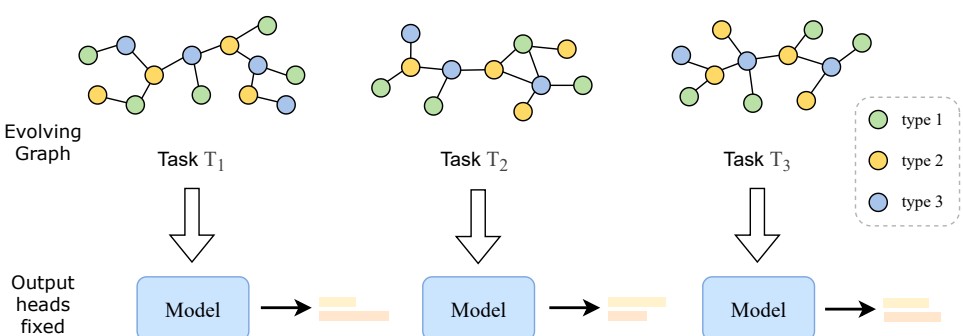

*Figure 5.* Illustration of the domain incremental setting.

## B.2. Baseline Introduction

We introduce the baseline models that we choose to compare in the following section:

**Elastic weight consolidation (EWC) (Kirkpatrick et al., 2017)** is a technique that regularizes the loss function, such that the model is encouraged to only modify the weights that are less important for the previous tasks. This is achieved by penalizing changes to weights that have large importance for the previously learned tasks, thereby mitigating catastrophic forgetting when learning new tasks.

**Memory aware synapses (MAS) (Aljundi et al., 2018)** is a regularization-based method that measures the importance of parameters based on the sensitivity of predictions to these parameters. When learning a new task, the model adjusts the weights according to the network's activations to update the important parameters relevant to the new task data.

**Topology-aware weight preserving(TWP) (Liu et al., 2021)** is a method introduced for graph continual learning that incorporates weight preservation mechanisms based on graph topology. By considering the local structure of the graph, TWP aims to overcome catastrophic forgetting. After computing the loss, the importance scores of the model weights are calculated according to the topology of the given graph, and the loss is regularized accordingly. This method leverages the properties of the graph.

**ER-GNN (Zhou & Cao, 2021)** stores samples from previous tasks as experiences and replays them when learning new tasks. After learning task $T_i$, sample nodes are saved to the buffer using a selection function. When learning the next task $T_{i+1}$, separate graphs are constructed for each learned task $\{T_t\}_{t=1,...,i}$. Then, the GNN is trained using these graphs. The overall loss is calculated by regularizing the current task's loss with the loss from the separately constructed graphs through experience replay.

**MetaCLGraph (Unal et al., 2023)** combines experience replay and meta-learning. In this method, the initial parameters of the model are calculated using the current task data. When learning a new task, stored samples from previous tasks are merged with the current task data to form a new graph, which is then used to update the model parameters.

**FTF-ER (Pang et al., 2024)** combines feature and global topological information by normalizing and weightedly integrating two types of node importance scores to evaluate node importance. Specifically, it introduces the Hodge Potential Score (HPS) module to capture global topological information. When learning a new task, a subgraph is induced using all the experience nodes stored in the buffer, the overall loss is calculated as the sum of the loss for the current task and the loss of the subgraph.

**DMSG (Qiao et al., 2025)** selects representative nodes using a diversity-aware buffer strategy that accounts for both intra-class and inter-class variations, and further augments replay through variationally generated synthetic embeddings. Adversarial and reconstruction-based objectives are used to preserve embedding quality and improve generalization during replay.

## B.3. Experimental Setting

Domain-Incremental Learning (Domain-IL) refers to a continual learning scenario where the task objective remains the same across tasks, but the data distribution (domain) shifts over time. This implies that the semantic meaning of the model's output stays fixed. For example, in knowledge graph completion, each task may involve different sets of entities and relations, but

the prediction goal—completing triplets—remains unchanged. Similarly, temporal data splits can form Domain-IL settings, where data from different time periods vary in distribution, yet the learning objective stays consistent.

### B.4. Implementation Detail

To evaluate the effectiveness and generalizability of our method, we conduct experiments on three HGNNs backbones: HAN (Wang et al., 2019), MAGNN (Fu et al., 2020), and HGT (Hu et al., 2020). Adam optimizer is used and the initial learning rate is set to 0.005 for all datasets. For HAN, each task is trained for 200 epochs with an early stopping patience of 100. For MAGNN, each task is trained for 100 epochs, with the early stopping patience set to 5 on DBLP and 10 on IMDB. For HGT, all tasks are trained for 300 epochs with a patience of 30. For datasets trained with mini-batches, the batch size is set to 8.

**Adaption for existing CGL methods:** Existing continual graph learning methods cannot be directly applied to our setting, particularly memory-replay–based approaches. Most replay strategies are designed for homogeneous graphs or assume a single target node type, which makes them incompatible with heterogeneous graphs where multiple node types and relations must be jointly preserved. To ensure a fair and meaningful comparison, we follow prior work by using their proposed strategies (or CM) to sample nodes of the target type, while employing our method to sample nodes of other types. The buffer size is uniformly set to 50 for the target node type in experience replay baselines (e.g., HERO, ER-GNN, MetaCLGraph, and FTF-ER), and 20 under mini-batch settings. For non-target node types, the buffer size is set to 200. The hyperparameters for regularization-based baselines (e.g., EWC, MAS, and TWP) are set following the TWP official repository and the benchmark study (ZHANG et al., 2022).

Our meta-learning module is configured under a 10-shot setting (2-way or 3-way depending on the dataset). For our method's hyperparameters, the knowledge distillation loss weight $\lambda_{kd}$ is selected between 0.1 and 1.0 for different datasets, and the logit-level distillation weight $\lambda_{\text{logit}}$ is selected between 0.6 and 1.5. We generally set the experience replay loss weight $\lambda_{er}$ and the semantic-level distillation weight $\lambda_{\text{sem}}$ to 1.0 and 10.0, respectively. The temperature in knowledge distillation is globally set to 1.0. All experiments are repeated five times with random seeds on Nvidia RTX A4000 and RTX 4090D GPUs, and we report the mean and standard deviation across all methods and datasets.

## C. Detailed Scalability Analysis

The results in Table 4 show that the regularization-based method TWP is clearly more efficient than replay-based methods in training, highlighting an open problem that we aim to address in future work. Nevertheless, our method significantly outperforms other replay-based approaches, demonstrating substantial improvements in computational efficiency over traditional replay-based methods. Although TWP is slightly more efficient than HERO, it consistently underperforms in terms of accuracy and forgetting across all benchmarks. Considering HERO's strong empirical performance, we argue that HERO provides a practical balance between efficiency and effectiveness, which can also be effectively trained on large-scale and long-sequence HCGL tasks.

*Table 4.* Training time (s) comparison on four datasets. Lower is better. Numbers in parentheses denote the ratio relative to HERO.

| Method | DBLP | IMDB | Freebase | Yelp |
|---|---|---|---|---|
| TWP | 0.157 (x0.42) | 0.032 (x0.78) | 0.068 (x0.59) | 3.020 (x0.50) |
| FTF-ER | 1.619 (x4.39) | 0.238 (x5.78) | 0.351 (x3.06) | 8.424 (x1.39) |
| ER-GNN | 0.411 (x1.11) | 0.053 (x1.29) | 0.078 (x0.68) | 6.652 (x1.10) |
| MetaCLGraph | 1.770 (x4.80) | 0.310 (x7.53) | 0.614 (x5.34) | 9.752 (x1.61) |
| HERO | 0.369 (x1.00) | 0.041 (x1.00) | 0.115 (x1.00) | 6.043 (x1.00) |

We provide detailed inference time results in Table 5. We observe that HERO achieves higher efficiency than most baseline methods during the inference phase, further supporting its practicality in real-world deployment.

## D. Additional Results

### D.1. Reordered Sequence

We further evaluate robustness under task-order variations by training with randomly permuted task sequences on DBLP, IMDB, and Freebase. Results is reported in Table 6 HERO continues to achieve the best overall performance under these randomized orders, indicating that its effectiveness does not depend on a specific task sequence. These results provide additional evidence of HERO's robustness and its ability to maintain stable performance under both structural perturbations

*Table 5.* Inference time (s) comparison on four datasets. HERO show the absolute inference time (s), while other methods are reported as relative difference (%). Positive difference means slower than HERO.

| Method | DBLP | IMDB | Freebase | Yelp |
|---|---|---|---|---|
| TWP | +1.99% | +3.81% | +2.22% | -1.54% |
| FTF-ER | +1.85% | +2.98% | +2.48% | +0.01% |
| ER-GNN | +1.02% | +2.78% | -1.13% | -2.13% |
| MetaCLGraph | +0.13% | +3.54% | +4.92% | +0.80% |
| HERO (Ours) | 3.4124 | 0.9815 | 2.5918 | 9.5556 |

and task-order uncertainty.

*Table 6.* Performance comparison on reordered task sequence. AP denotes average performance (higher is better) and AF denotes average forgetting (lower is better).

| Methods | DBLP | | IMDB | | Freebase | |
|---|---|---|---|---|---|---|
| | AP↑ | AF↓ | AP↑ | AF↓ | AP↑ | AF↓ |
| Finetune | 83.5 | 24.3 | 56.7 | 49.8 | 49.2 | 31.8 |
| EWC | 90.7 | 10.4 | 67.8 | 24.0 | 54.0 | 24.2 |
| MAS | 92.6 | 4.8 | 72.0 | 13.0 | 47.1 | 34.6 |
| TWP | 93.6 | 3.5 | 73.4 | 7.6 | 52.0 | 26.4 |
| ER-GNN | 93.2 | 6.1 | 69.7 | 13.6 | 56.0 | 17.7 |
| FTF-ER | 94.6 | 1.1 | 67.2 | 30.4 | 45.1 | 35.6 |
| MetaCLGraph | 94.7 | 3.4 | 65.6 | 26.9 | 50.1 | 26.2 |
| Ours | **95.6** | **1.8** | **75.5** | **8.0** | **61.0** | **5.1** |

## D.2. Metapath

Here we study the effect of meta-path availability on model performance by randomly removing one or two meta-paths during training. The results, reported in Table 7, show that all methods experience performance degradation as meta-path information is reduced. Nevertheless, HERO consistently maintains the strongest performance across all settings. This behavior is expected, as the backbone itself relies on meta-paths to encode semantic structure; removing meta-paths therefore weakens the representational capacity of the underlying model and leads to an inherent performance drop. Despite this limitation, HERO demonstrates superior robustness to meta-path reduction compared to competing methods.

## E. Additional Ablation Study

We further analyze the contribution of the two submodules in the HKD module—logit-level distillation and semantic-level distillation—to overall model performance, and assess the effectiveness of our proposed design.

Figure 7 illustrates the impact of different distillation strategies—node-level only ("with node"), semantic-level only ("with sem"), and the combination of both ("with both")—on the model's Average Performance (AP) and Average Forgetting (AF) across four datasets (DBLP, IMDB, Freebase, Yelp). The results show that the combined strategy ("with both") consistently achieves the highest AP and lowest AF across all datasets, demonstrating that logit-level and semantic-level distillation complement each other. Semantic-level distillation alone ("with sem") exhibits weaker forgetting suppression on Freebase and Yelp, suggesting it may struggle to independently capture task-level semantic changes. On complex heterogeneous graphs such as Yelp, the synergy of both distillation levels yields especially significant performance improvements, indicating that addressing both structural and semantic forgetting is crucial.

## F. Extended Hyperparameters Analysis

Figure 8 shows the changes in AP (left) and AF (right) across datasets under varying logit-level distillation loss weights (x-axis). For most datasets, AP remains relatively stable, indicating robustness to this hyperparameter. IMDB and Yelp exhibit more fluctuation, likely due to significant distribution shifts across tasks, making them more sensitive to weight

*Table 7.* Performance comparison under two settings: removing one metapath (Setting 1) and removing two metapaths (Setting 2). AP denotes average performance (↑) and AF denotes average forgetting (↓).

| Setting | Method | DBLP | | IMDB | | Freebase | |
|---|---|---|---|---|---|---|---|
| | | AP ↑ | AF ↓ | AP ↑ | AF ↓ | AP ↑ | AF ↓ |
| Setting 1 | Finetune | 82.3 | 26.8 | 67.8 | 27.0 | 51.3 | 28.8 |
| | EWC | 85.0 | 21.9 | 69.1 | 26.6 | 55.2 | 21.8 |
| | MAS | 90.6 | 8.9 | 71.0 | 21.2 | 55.7 | 18.1 |
| | TWP | 90.0 | 10.4 | 74.3 | 15.3 | **58.3** | **14.1** |
| | ER-GNN | 89.4 | 13.4 | 74.1 | 13.5 | 53.1 | 22.7 |
| | FTF-ER | 90.2 | 10.2 | 71.5 | 20.6 | 54.9 | 17.8 |
| | MetaCLGraph | 89.6 | 11.8 | 74.1 | 15.5 | 55.9 | 16.3 |
| | Ours | **93.2** | **4.8** | **77.8** | **8.0** | 58.2 | 14.1 |
| | Joint Train | 94.8 | 2.1 | 79.5 | 4.7 | 62.6 | 6.1 |
| Setting 2 | Finetune | 80.5 | 28.5 | 67.6 | 26.8 | 50.9 | 28.6 |
| | EWC | 82.9 | 23.4 | 68.8 | 26.8 | 55.1 | 21.9 |
| | MAS | 88.9 | 10.1 | 71.1 | 21.7 | 55.9 | 18.0 |
| | TWP | 88.1 | 12.5 | 73.6 | 15.8 | 57.2 | **14.2** |
| | ER-GNN | 87.5 | 14.9 | 73.6 | 14.1 | 53.6 | 23.3 |
| | FTF-ER | 88.8 | 11.6 | 70.8 | 20.7 | 54.5 | 17.6 |
| | MetaCLGraph | 87.7 | 13.3 | 74.0 | 16.3 | 56.5 | 16.2 |
| | Ours | **91.3** | **6.2** | **77.6** | **8.5** | **58.0** | 14.7 |
| | Joint Train | 93.0 | 4.0 | 79.1 | 5.1 | 61.9 | 6.3 |

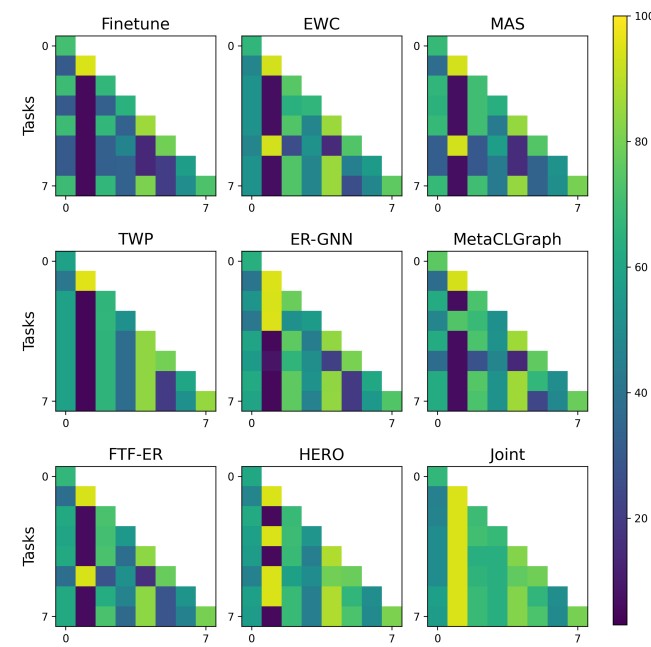

*Figure 6.* Visualization: Accuracy matrices on the Yelp dataset under 2-way setting.

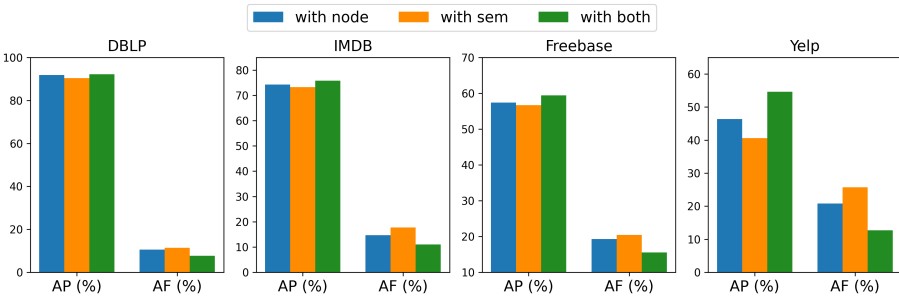

*Figure 7.* Average performance (AP) and average forgetting (AF) of the HERO method under three settings: with logit-level distillation, with semantic-level distillation, and with both.

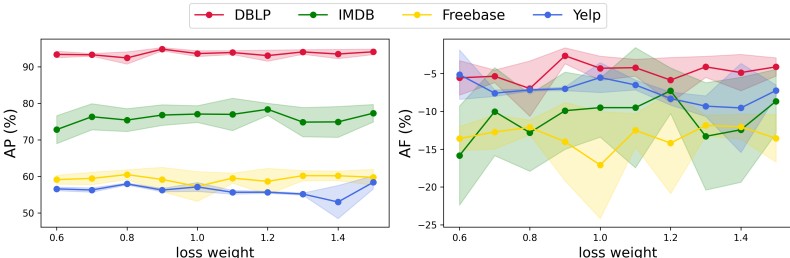

*Figure 8.* Sensitivity analysis of the logit-level distillation weight $\lambda_{logit}$ in the HKD module. Left: Average Performance (AP); Right: Average Forgetting (AF).

changes. The AF curves reveal that moderate weights (0.8 to 1.2) generally result in lower forgetting, while too high or too low weights cause an imbalance between new and old task focus. Yelp shows the smallest AF variation, suggesting that the HKD mechanism performs steadily in complex heterogeneous settings, aiding the model's generalization ability.

## G. Forgetting-aware Gap

AP (Average Performance) measures the mean test accuracy across all previously learned tasks, while AF (Average Forgetting) quantifies the performance drop on a specific task after learning subsequent ones. However, we observe that in order to retain knowledge from earlier tasks, the model often compromises its performance on the final task. To assess this trade-off, we introduce a new metric called Forgetting-aware Gap (FaG), defined as the difference between the test accuracy obtained by training the model solely on the final task and the test accuracy on the same task after completing all tasks under the continual learning framework:

$$\text{FaG} = a_T^{FT} - a_T^{CL} \tag{18}$$

where $a_T$ denotes the test accuracy on the final task $\mathcal{T}_T$, with "FT" referring to the finetuning baseline (i.e., training only on task $\mathcal{T}_T$), and "CL" referring to the accuracy obtained after training with a continual learning method. This gap reflects the performance degradation caused by the inherent plasticity-stability trade-off in continual learning.

*Table 8.* Forgetting-aware Gap (FaG) comparison on four datasets with HAN as the backbone model (averaged over 5 runs). Lower is better.

| Method | DBLP | IMDB | Freebase | Yelp |
|---|---|---|---|---|
| Joint Train | 0.69 | 2.47 | 11.59 | 11.58 |
| EWC | 2.15 | 2.78 | 10.17 | 1.62 |
| MAS | 0.40 | 0.97 | 9.53 | 2.09 |
| TWP | 1.03 | 2.27 | 10.44 | 5.58 |
| ER-GNN | 0.22 | 2.62 | 8.44 | 8.46 |
| MetaCLGraph | 1.36 | 2.87 | 13.68 | 13.61 |
| FTF-ER | 0.83 | 3.43 | 10.59 | 7.19 |
| HERO (Ours) | 0.63 | 1.31 | 9.90 | 6.88 |

**Results Analysis** MetaCLGraph and FTF-ER suffer high FaG values on Freebase and Yelp (13.68 and 10.59), reflecting performance degradation under distribution shifts—a manifestation of the stability-plasticity dilemma. ER-GNN achieves

the lowest FaG (0.22) on DBLP, showing good adaptability, but its FaG sharply rises to 8.46 on Yelp, indicating that simple replay mechanisms struggle with complex heterogeneous structures. In contrast, our HERO method consistently maintains lower FaG values. Notably, it achieves 1.31 on IMDB and 6.88 on Yelp, outperforming experience-replay and regularization-based methods like TWP (2.27 on IMDB). These results validate that HERO, through its meta-learning and heterogeneity-aware distillation design, effectively mitigates final-task degradation while preserving task adaptability. Overall, FaG serves as a useful metric to quantify the trade-off between knowledge retention and adaptation, further supporting the effectiveness of HERO in heterogeneous continual learning.

