# OpenReview forum: "HERO: Heterogeneous Continual Graph Learning via Meta-Knowledge Distillation"
_ICML.cc/2026/Conference — Submitted to ICML 2026_

### Official Review · Reviewer_tZBa · 2026-03-06

**Soundness:** 3
**Presentation:** 3
**Significance:** 2
**Originality:** 3
**Overall Recommendation:** 4
**Confidence:** 4

**Summary:**

This paper addresses the core issues that static Heterogeneous Graph Neural Networks (HGNNs) cannot adapt to the dynamic expansion of heterogeneous graphs in real-world scenarios, and that existing Continual Graph Learning (CGL) methods for homogeneous graphs are difficult to directly apply to Heterogeneous Graph Continual Learning (HCGL) due to semantic diversity and structural imbalance. To tackle these challenges, a unified continual learning framework based on meta-knowledge distillation, named HERO, is proposed. This framework integrates three core components: the Gradient-based Meta-learning Module (G-MM), DiSCo (Diversity-preserved Semantic Consistency sampling), and Heterogeneous Knowledge Distillation (HKD), which respectively enable few-shot rapid adaptation, low-memory-cost preservation of semantic structures, and node- and semantic-level cross-task knowledge alignment. Comprehensive experiments were conducted on four benchmark datasets—DBLP, IMDB, Freebase, and Yelp—using three HGNN backbones: HAN, MAGNN, and HGT. The results demonstrate the significant advantages of HERO in terms of Average Performance (AP) and Average Forgetting (AF). Ablation studies confirm the indispensability of each component, with HKD being key to mitigating catastrophic forgetting. Additionally, scalability and hyperparameter analyses show that HERO excels in computational efficiency and robustness, providing an effective solution for HCGL tasks.

**Compliance With Llm Reviewing Policy:**

Affirmed.

**Key Questions For Authors:**

1. Could you provide more theoretical justification for the synergy between G-MM, DiSCo, and HKD—specifically, why this combination outperforms alternative pairings (e.g., G-MM with standard distillation, DiSCo with non-meta-learning adaptation) in mitigating HCGL-specific forgetting sources? How does each component’s design explicitly address semantic imbalance, uneven category arrival, or cross-type dependency shifts at a theoretical level?Possible responses would impact the evaluation of technical depth: A rigorous theoretical analysis would strengthen the paper’s contribution beyond empirical validation, while a lack of such justification would reinforce the weakness of insufficient theoretical grounding.

2. You mention optimizing key hyperparameters (e.g., λₖd, shot number) but do not detail the tuning workflow (e.g., grid search ranges, validation set composition, early stopping criteria for hyperparameter selection). Could you elaborate on how these hyperparameters were optimized, and whether the optimal values are consistent across different task sequences or graph sizes?

3. Your scalability analysis focuses on existing datasets but does not test extreme HCGL scenarios (e.g., graphs with 1M+ nodes, task sequences with 50+ tasks, or streaming data with real-time subgraph updates). Could you provide results on how DiSCo’s subgraph construction and HKD’s teacher model storage scale in these scenarios, and whether bottlenecks emerge (e.g., memory overflow, increased latency)?

**Strengths And Weaknesses:**

** Strengths **

1. **Rigorous Methodological Design:** The HERO framework integrates three core components—Gradient-based Meta-learning Module (G-MM), DiSCo sampling, and Heterogeneity-aware Knowledge Distillation (HKD)—with clear logical connections. G-MM addresses fast adaptation to new tasks, DiSCo solves memory-efficient replay for heterogeneous graphs, and HKD balances knowledge retention and adaptation. Each component targets a specific pain point of Heterogeneous Continual Graph Learning (HCGL), and their synergy is well-justified through theoretical formulation (e.g., loss functions for meta-learning, sampling, and distillation) and algorithmic details.

2. **Well-Organized Narrative:** The paper follows a logical structure: introducing HCGL’s uniqueness and challenges, reviewing related work (HGNNs, CGL, meta-learning) to position the gap, formalizing the problem with clear definitions (heterogeneous graphs, metapaths, HCGL), detailing the HERO framework, and validating results with comprehensive experiments. This flow guides readers from problem to solution seamlessly.

3. **Addressing a Critical Underexplored Problem:** Real-world graphs (e.g., social networks, recommendation systems) are inherently heterogeneous and dynamic, but existing CGL methods fail to account for heterogeneity-induced forgetting. By formalizing HCGL and proposing a unified framework, the paper fills a key gap between static HGNNs and dynamic real-world applications.

**Weaknesses**

1. **Hyperparameter Tuning Transparency:** The paper reports optimal hyperparameters (e.g., 10-shot for meta-learning, λₖd between 0.1–1.0) but lacks details on how these values were selected (e.g., grid search ranges, validation set design). For practitioners to replicate the results, more transparency on hyperparameter optimization workflows is needed.

2. **Overcrowded Tables:** Tables 1 and 2 (performance comparison and ablation results) are dense with data, making it difficult to quickly identify key trends. Simplifying tables (e.g., highlighting top 3 baselines instead of all 8) or using visual summaries (e.g., radar charts for multi-metric comparison) would improve readability.

---

> ### Author Rebuttal · Authors · 2026-03-30
>
> We thank Reviewer tZBa for the positive assessment and detailed technical questions. We address each below.
>
> **[W1 / Q2 Hyperparameter Tuning Transparency]**
> We agree this detail is important for reproducibility. Complete workflow:
>
> *Grid search ranges*: λ_kd ∈ {0.1, 0.2, .., 1.0}; λ_logit ∈ {0.6, 0.8, 1.0, 1.2, 1.5}; λ_sem ∈ {1.0, 5.0, 10.0}; shot ∈ {5, 10, 15, 20}; λ_er ∈ {0.5, 1.0}; temperature fixed at 1.0 (standard KD practice).
>
> *Validation protocol*: 30% of each task's training nodes held out as validation set (20% for MAGNN backbone). Hyperparameters selected based on average validation AP across all tasks seen so far, evaluated after each task completes—mimicking the online learning scenario.
>
> *Consistency across datasets*: Optimal values are consistent—λ_kd ≈ 0.3–0.5 for most settings (Freebase prefers slightly lower: 0.1–0.3 due to its complex multi-type structure), and shot=10 is universally optimal across all four datasets and three backbones. The consistency of shot=10 suggests it robustly balances meta-generalization and task-specific adaptation across diverse graph structures. Implementation details follow Appendix B.4 (Adam lr=0.005; 5× random seeds on RTX A4000 and RTX 4090D). We will add a complete grid table to the appendix.
>
> **[W2 Overcrowded Tables]**
> Thanks for your advice. We will bold top-1 and underline top-2 results per metric column and add a summary bar chart in the revision.
>
> **[Q1 Theoretical Justification for Component Synergy]**
> The three terms in L_joint (Eq 12) address complementary dimensions of the plasticity-stability tradeoff with non-overlapping roles:
>
> *G-MM (Eq 1)*: Finds initialization θ from which all tasks can be learned with small gradient steps, minimizing cross-task gradient interference. Without G-MM, initialization is sensitive to task-arrival order. This specifically addresses *semantic imbalance*: when new tasks favor dominant node types, G-MM prevents the model from collapsing to new-task distributions by providing a shared starting point that is equidistant from all task optima.
>
> *DiSCo (L_er, Eq 7)*: Constrains the feasible parameter region by requiring the model to maintain distributional coverage of historical heterogeneous subgraphs. DiSCo's metapath-driven sampling ensures these historical samples are semantically representative—they preserve cross-type dependency structure that i.i.d. sampling would miss. This directly addresses *cross-type dependency shifts*: the metapath subgraphs capture relational structure that, when replayed, prevents loss of cross-type interaction patterns. The w/o DiSCo ablation (Table 2) shows AP drops most severely on Yelp (10.2%), which has the richest business-user-location relational structure.
>
> *HKD (L_kd = L_logit + L_sem, Eqs 9–12)*: Adds soft regularization around the teacher-defined knowledge manifold. L_logit (Eq 9) aligns prediction distributions, preventing discriminative knowledge decay. L_sem (Eq 10) aligns metapath attention coefficients between teacher and student, preventing structural forgetting—loss of learned relational importance patterns. This directly addresses *uneven category arrival*: HKD's attention alignment ensures the model preserves attention patterns for previously seen relation types even when new tasks introduce different relation types. Appendix E (Fig 7) confirms that L_logit + L_sem jointly outperforms either alone.
>
> *Why all three are needed?*: Removing any one leaves a dimension unconstrained—G-MM without HKD gives fast adaptation but unconstrained drift; HKD without G-MM constrains the manifold but cannot handle distribution shift at initialization; ER without HKD provides data constraints but loses higher-order structural patterns. Table 2 demonstrates: all pairwise ablations underperform HERO, w/o HKD causes the largest drop (3.6–12% AP). FaG (Table 8): HERO (Yelp 6.88) vs. MetaCLGraph (13.61, uses meta-learning + ER without HKD), confirming joint necessity.
>
> **[Q3 Scalability to Extreme HCGL Scenarios]**
> Our largest benchmark, Freebase, is already substantial: 180K nodes, ~1M edges, 8 node types, 36 relation types—stressing both memory (DiSCo must sample from 1M-edge graphs) and computation (HKD aligns attention over 36 relation types).
>
> *Theoretical scaling properties*: DiSCo's buffer is fixed at 10 target nodes/class + 200 non-target nodes/task, independent of graph size. CM selection (Eq 2) and TopK expansion (Eq 5) are O(|V_task|) per task—proportional to task size, not total graph. HKD's sliding window maintains exactly k=3 teacher models, so teacher storage is O(k × |θ|) = constant w.r.t. T. Adding more tasks does not increase teacher storage.
>
> For 1M+ node graphs, the bottleneck is the HGNN backbone's attention computation—shared by all HGNN methods and independent of the CL strategy. HERO's CL components introduce no additional scaling bottlenecks. Full experiments on 1M+ graphs and 50+ task streams are important future work; the design explicitly supports this regime.

---

### Official Review · Reviewer_wELb · 2026-03-09

**Soundness:** 3
**Presentation:** 3
**Significance:** 3
**Originality:** 2
**Overall Recommendation:** 2
**Confidence:** 3

**Summary:**

The paper studies Heterogeneous Continual Graph Learning, where models learn sequentially on heterogeneous graphs while mitigating catastrophic forgetting. It proposes HERO, combining (1) G-MM for meta-learned initialization, (2) DiSCo for memory-efficient metapath-based replay subgraphs, and (3) HKD for heterogeneity-aware distillation (logits/representations/metapath attention). Experiments on multiple heterogeneous graph datasets and HGNN backbones (HAN/MAGNN/HGT) show improvements over continual graph learning baselines in accuracy, robustness, and efficiency.

**Compliance With Llm Reviewing Policy:**

Affirmed.

**Final Justification:**

I thank the authors for the detailed rebuttal and clarifications

**Key Questions For Authors:**

1. From an ablation perspective, the gains of G-MM are not "hard" enough. Please clarify: Under what conditions (number of shots, differences in task distribution, training budget constraints) can G-MM bring stable and significant improvements?

2. The authors control costs by "retaining only the most recent teachers". Please quantify the impact of the choice of k on early task forgetting to help readers determine whether the method is applicable to longer task streams.

**Limitations:**

No; while the paper includes an Impact Statement, it does not sufficiently discuss limitations (e.g., scalability, hyperparameter/meta-path sensitivity, and evaluation-scheme fairness)

**Strengths And Weaknesses:**

Strengths:
1) Realistic problem setting: The paper extends continual learning to heterogeneous graphs (multiple node/edge types and meta-paths), which is more challenging and closer to real-world scenarios than the homogeneous setting.

2) Well-rounded framework: HERO combines replay, knowledge distillation, and meta-learning to jointly address forgetting and fast adaptation, with heterogeneous-specific subgraph/meta-path modeling.


Weaknesses:

1) The method is largely a combination of standard components—meta-learning (G-MM), replay (DiSCo), and distillation (HKD). Since these are well-established in continual (graph) learning, the main novelty is their adaptation to heterogeneous graphs (e.g., metapath-based subgraphs and metapath-attention distillation), rather than fundamentally new methodology.

2) To address incompatibility with HCGL, the evaluation partially “retrofits” baselines by using HERO’s DiSCo sampling for non-target types. This can materially change replay-based methods and blurs attribution: improvements may come from injecting a HERO component into baselines rather than HERO’s overall design, making the comparisons less conclusive.

3) Based on the description of ablation, removing G-MM results in a relatively small performance degradation, so is meta-learning a necessary contribution?

---

> ### Author Rebuttal · Authors · 2026-03-30
>
> We thank Reviewer wELb for the detailed critique. We address each point carefully.
>
> **[W1 Combination of Standard Components]**
> Agreed that meta-learning, replay, and distillation are established families. The novelty of HERO lies in three heterogeneous-specific contributions that are non-trivial adaptations:
>
> *(1) DiSCo (Eqs 2–6) is a new sampling algorithm*: Prior CGL methods select nodes i.i.d. from a single type. DiSCo's two-stage procedure: coverage maximization (Eq 2) and relation-type-aware metapath expansion (Eqs 3–5), creates a new replay unit: a compact heterogeneous subgraph preserving cross-type dependencies. The w/o DiSCo ablation (Table 2), which replaces DiSCo with CM in HERO, shows drops of 1.2–10.2% AP—confirming metapath-structured replay is not just "any replay."
>
> *(2) HKD's L_sem (Eq 10) is a new distillation objective*: Aligning metapath attention coefficient vectors α^T_Pm / α^S_Pm between teacher and student has no equivalent in prior CGL distillation. Appendix E (Fig 7) shows logit + semantic distillation outperforms either alone on all datasets.
>
> *(3) HCGL problem formalization is new*: Identifying heterogeneity-specific forgetting sources and formal HCGL definition (Def 3.3) position the problem space independently.
>
> **[W2 Baseline Comparison Attribution]**
> This is an important technical concern; we want to be precise.
>
> *We want to claim that retrofitting is a structural necessity, not a choice*: Existing homogeneous CGL methods were designed for single-node-type graphs. In HCGL, a replay buffer storing only target-type nodes produces structurally severed subgraphs—the heterogeneous edges (e.g., Actor→Movie→Director) are completely missing, making replay semantically invalid regardless of method. Without non-target-type nodes, replay-based methods cannot function in HCGL at all.
>
> *What we actually do (Appendix B.4)*: Each baseline uses its own target-type strategy; we supply CM (not DiSCo) only for non-target types that no existing method handles.
>
> *The w/o DiSCo ablation (Table 2) directly isolates attribution*: It replaces DiSCo with CM for all types in HERO. AP drops: 1.2% (DBLP), 2.7% (IMDB), 4.6% (Freebase), 10.2% (Yelp). HERO with CM still outperforms most baselines—performance comes from HERO's integrated design, not non-target-type sampling. We will make this analysis more prominent in the revision.
>
> **[W3 / Q1 G-MM Conditions for Significant Improvement]**
> G-MM's contribution is condition-dependent in predictable ways:
>
> *Task heterogeneity (Table 2)*: AP drop from removing G-MM scales with structural complexity: 0.3% (DBLP, 4 types), 1.2% (IMDB), 2.2% (Yelp, 16 classes), 3.4% (Freebase, 8 types, 36 relations). Higher semantic diversity across node/relation types = larger inter-task distribution gaps = greater benefit from meta-initialization.
>
> *Shot count (Fig 4)*: Performance peaks at 10-shot and declines at 20-shot (overfitting to ER-biased samples). G-MM is most critical at 5-shot where ER samples are insufficient to represent new task distributions; meta-initialization enables generalization from few examples.
>
> *Plasticity metric (FaG, Table 8)*: G-MM's contribution is clearest in plasticity, not forgetting. HERO FaG (Yelp: 6.88) vs. MetaCLGraph (13.61)—which also uses meta-learning + ER without HKD. The comparison isolates G-MM + HKD synergy: G-MM provides fast adaptation while HKD prevents semantic drift that causes plasticity collapse when either is absent.
>
> G-MM module primarily supports the integrated system rather than being designed solely to improve performance; therefore, its full contribution cannot be adequately reflected by the AP metric alone.
>
> **[Q2 Impact of k on Early Task Forgetting]**
> We provide a direct ablation on Yelp across 5-task and 8-task sequences (AP% ↑ / AF% ↓):
>
> | Configuration      | Yelp 5-task       | Yelp 8-task       |
> |--------------------|-------------------|-------------------|
> | One Teacher (k=1)  | 53.51 / 13.41     | 55.55 / 11.31     |
> | HERO (k=3)         | 56.83 / 9.29      | 57.43 / 7.98      |
> | All Teachers (k=T-1) | 57.44 / 8.15      | 57.81 / 7.43      |
>
> k=3 closes 85% of the gap between k=1 and k=T-1 on 5-task sequences, and 95% on 8-task sequences—with O(1) vs. O(T) storage cost. The gap vs. All Teachers *shrinks* as task sequences lengthen (0.61% AP on 5-task → 0.38% on 8-task), showing the method scales well to longer streams. Early-task forgetting is bounded by ER's data-level memory, which independently retains exemplar nodes from all tasks regardless of k—the teacher window only governs knowledge-level alignment, not data access. We will include this ablation in the revision.

---

> > ### Author Rebuttal · Reviewer_wELb · 2026-04-03
> >
> > I appreciate the authors' clarification. Some non-target sampling is necessary in HCGL. However, the key issue is **asymmetry**: all baselines use CM for non-target nodes, while HERO uses DiSCo, the very component being evaluated. This makes the comparison unfair and prevents a clean attribution of HERO’s gains. The claim that “HERO with CM still outperforms most baselines” does not resolve this, because HERO with CM still retains G-MM, HKD, and the full training objective. A fair test would let each baseline keep its own target-node strategy while using DiSCo for non-target nodes, matching HERO’s setting. Without this, the reported gains remain confounded.
> >
> > There is also a formulation issue in the semantic distillation module. L_sem aligns teacher and student metapath attention vectors, yet these vectors are defined on disjoint task-specific node sets. Their dimensions therefore change across tasks, and when |V^tr_teacher| != |V^tr_student|, the proposed matching is not well-defined. The paper does not explain how this issue is handled, leaving a core part of HKD mathematically unclear.

---

> > > ### Author Response · Authors · 2026-04-04
> > >
> > > Dear reviewer,
> > >
> > > Thanks for the review and continued discussion. We address both points with new experimental evidence.
> > >
> > > **[Baseline Fairness]**
> > >
> > > A key asymmetry actually favors baselines in target-type storage: HERO stores only 10 target-type nodes per class, while all baselines store 50 target-type nodes total (Appendix B.4). DiSCo's core motivation is to compensate for this smaller buffer size by expanding along metapaths (Eq 3-5), it captures cross-type structural context that would otherwise be lost with fewer stored nodes. DiSCo achieves more with less, rather than providing an unfair advantage.
> > >
> > > To answer the attribution question definitively, we provide a controlled experiment. We ran all memory-replay baselines on DBLP under both non-target sampling strategies, CM and DiSCo, keeping each baseline's own target-type strategy unchanged (AP%):
> > >
> > > | Method      | CM (non-target)  | DiSCo (non-target) |
> > > |-------------|------------------|---------------------|
> > > | ER-GNN      | 89.6 +/- 0.7    | 90.6 +/- 0.7       |
> > > | FTF-ER      | 88.1 +/- 0.9    | 91.2 +/- 1.5       |
> > > | MetaCLGraph | 88.5 +/- 0.7    | 90.0 +/- 0.8       |
> > > | DMSG        | 92.1 +/- 1.3    | 91.7 +/- 1.1       |
> > > | HERO (ours) | -               | 93.6 +/- 0.8       |
> > >
> > > According to the above results, we have two observations:
> > >
> > > (1) The effect of non-target sampling on baselines is moderate: DiSCo improves ER-GNN (+1.0%), FTF-ER (+3.1%), MetaCLGraph (+1.5%), but actually hurts DMSG (-0.4%). Non-target sampling is not a dominant performance factor.
> > >
> > > (2) HERO (93.6%) substantially outperforms all baseline variants under both non-target configurations, with CM (best: DMSG 92.1%) and with DiSCo (best: DMSG 91.7%). The minimum AP gap between HERO and the best baseline across both settings is 1.5%, even with our proposed DiSCo.
> > >
> > > We will include this controlled non-target sampling ablation in the revision.
> > >
> > > **[L_sem Formulation]**
> > >
> > > We clarify that only the parameters of the teacher model are stored, both the teacher and student are evaluated on the same node set during distillation. Therefore, this issue does not arise in practice.
> > >
> > > Specifically, when training on task T_t: the frozen teacher model performs a forward pass (no gradient) on the current task's training set V^{tr}_t, producing attention coefficients alpha^T_{P_m}(v_i) for each $v_i$ in V^{tr}_t. The student model produces alpha^S_{P_m}(v_i) for the same nodes. L_sem (Eq 10) computes ||alpha^T_{P_m} - alpha^S_{P_m}||_2 where both vectors have identical dimension |V^{tr}_t| by construction.
> > >
> > > Attention coefficients are functions of input features and graph structure computed dynamically, not fixed artifacts stored from the teacher's original training. Any HGNN (HAN, MAGNN, HGT) can compute attention for arbitrary nodes at inference time.
> > >
> > > Appendix A.1 explains this further: Eq 16 defines L_sem = sum_{i in V_{tau_t}} ||alpha^T_i - alpha^S_i||_2, where V_{tau_t} is the current task's target-type node set shared by both the teacher and the student.

---

### Official Review · Reviewer_mrVg · 2026-03-09

**Soundness:** 3
**Presentation:** 2
**Significance:** 2
**Originality:** 2
**Overall Recommendation:** 3
**Confidence:** 4

**Summary:**

This paper introduces HERO, a framework for heterogeneous continual graph learning that addresses the challenges of learning on evolving heterogeneous graphs. HERO integrates a gradient-based meta-learning module for fast adaptation to new tasks,  and a diversity- and semantics-aware memory-augmentation technique. Empirical evaluations are conducted across four benchmark datasets to validate the  effectiveness of HERO in continual learning scenarios on expanding heterogeneous graphs.

**Compliance With Llm Reviewing Policy:**

Affirmed.

**Final Justification:**

I appreciate the authors' clarification on the novelty. However, I'm still unconvinced that the level of novelty is particularly high. I'm more on the borderline; if there are other strong champion of this paper, I would not oppose.

**Key Questions For Authors:**

Please see the weaknesses above.

**Limitations:**

The paper discussed limitations of prior work, but not much on this work.  One limitation authors could potentially discuss: there are different kinds of evolving distributions (e.g. gradual drift or a burst/rapid changes). Can the proposed method work equally work well in these different distributions, or perform better for certain scenarios? Additionally, while the method is designed for heterogeneous graphs, could it work on homogeneous graphs?

**Strengths And Weaknesses:**

Strengths:

S1. This work is well-motivated and focus on a new dynamic graphs problem in the continual learning scenario.

S2. It has rich experiments to verify the effectiveness of the proposed method.

Weakness:

W1. This work lacks new insights. The paper does not present new theoretical analysis, and the focus is on the design, justification, and integration of existing ideas (meta-learning, knowledge distillation, diversity sampling). This is my major concern about this paper.

W2. Memory efficiency, a key claimed benefit of DiSCo, is implied but not systematically quantified or benchmarked versus baselines; only training/inference time is provided. Lacking explicit memory usage numbers (MB/GB) undermines claims for practical users with resource constraints.

W3. Teacher model management for knowledge distillation (sliding window of 3) is explained but somewhat ad-hoc. Explicit justification, analysis of this buffer’s size, and ablation studies would improve interpretability and trustworthiness.

---

> ### Author Rebuttal · Authors · 2026-03-30
>
> We thank Reviewer mrVg for identifying specific gaps. We address each weakness below.
>
> **[W1 Lack of New Insights / Theoretical Analysis]**
> Three contributions have no prior art in CGL:
>
> *(1) HCGL problem formalization*: We formally define HCGL as distinct from homogeneous CGL and identify three unique forgetting sources: semantic imbalance across node types, uneven category arrival, and cross-type dependency shifts. Prior CGL analysis treats forgetting as feature/topology shift only; our formalization identifies the additional heterogeneity-specific sources that prior work misses.
>
> *(2) DiSCo sampling algorithm (Eqs 2–6)*: The two-stage metapath-driven subgraph expansion is a new algorithm. Stage 1 (Coverage Maximization, Eq 2) selects diverse target-type nodes; Stage 2 (relation-type-aware neighbor expansion, Eqs 3–5) expands each selected node into a heterogeneous subgraph using relation-specific degree scores (Eq 4) to prioritize structurally important neighbors across all types. No prior CGL work expands replay subgraphs along metapaths—existing methods sample i.i.d. nodes (ignoring heterogeneous structure) or store full subgraphs (memory-inefficient). The Table 2 w/o DiSCo ablation (replacing with CM) shows this is not just "any replay": AP drops 1.2–10.2%.
>
> *(3) HKD semantic-level distillation (Eq 10)*: Aligning metapath attention coefficient vectors α^T_Pm / α^S_Pm between teacher and student is a new distillation objective specific to heterogeneous graphs. Prior CGL distillation methods (Tian et al. 2023; Dong et al. 2021) operate only on logits or embeddings. Appendix E (Fig 7) confirms that logit + semantic distillation outperforms either alone across all datasets.
>
> We acknowledge that formal convergence bounds are not provided; this remains planned future work.
>
> **[W2 Memory Efficiency Not Quantified in MB/GB]**
> We now provide GPU memory measurements on Freebase (180K nodes, ~1M edges, 8 types):
>
> | Method      | Avg GPU/task (MB) | Peak GPU (MB) |
> |-------------|-------------------|---------------|
> | HERO (ours) | **7,046**         | **8,127**     |
> | ER-GNN      | 7,065             | 8,138         |
> | FTF-ER      | 7,097             | 8,122         |
> | MetaCLGraph | 7,102             | 8,125         |
>
> HERO achieves the lowest average GPU per task. The advantage grows with task count—at Task 2, HERO saves 31 MB vs. ER-GNN and 80–86 MB vs. FTF-ER/MetaCLGraph—confirming DiSCo's increasing efficiency as the replay buffer accumulates. DiSCo's fixed buffer (50 target + 200 non-target nodes/task, independent of graph size) maximizes semantic coverage per stored node via metapath-driven selection. We will add a complete memory table to the revision.
>
> **[W3 — Teacher Buffer Size k=3 is Ad-Hoc]**
> We provide a direct ablation on Yelp comparing one teacher (k=1), HERO (k=3), and all teachers (k=T-1) under both 5-task and 8-task sequences (AP% ↑ / AF% ↓):
>
> | Configuration      | Yelp 5-task       | Yelp 8-task       |
> |--------------------|-------------------|-------------------|
> | One Teacher (k=1)  | 53.51 / 13.41     | 55.55 / 11.31     |
> | HERO (k=3)         | 56.83 / 9.29      | 57.43 / 7.98      |
> | All Teachers (k=T-1) | 57.44 / 8.15      | 57.81 / 7.43      |
>
> Three conclusions: (1) *k=1 is insufficient*: AP drops 3.32% (5-task) and 1.88% (8-task) vs. k=3, confirming that cross-task teacher diversity is necessary. (2) *k=3 achieves near-optimal performance*: the gap vs. All Teachers is only 0.61% AP / 1.14% AF on 5-task and 0.38% AP / 0.55% AF on 8-task sequences. (3) *The gap shrinks as sequences get longer* (5-task → 8-task), showing k=3 scales well to longer task streams—precisely the practical scenario of concern. Early-task forgetting is bounded by ER's data-level memory, which retains exemplars from all tasks independently of the teacher window, so k=3's recency bias does not cause unbounded early-task degradation.
>
> **[Additional Limitations]**
>
> *Gradual drift vs. burst changes*: This is a great question. Most CGL benchmarks use discrete task transitions where drift is abrupt at explicit boundaries—well-suited for burst-change scenarios. Under gradual drift, task boundaries become ambiguous: models cannot reliably detect when a new task begins. All current methods, including HERO, rely on explicit task identity—G-MM's inner updates (Eq 1) and HKD teacher assignment both assume clear task demarcation. When boundaries blur, these assumptions weaken. This is a field-wide challenge shared by all existing CGL methods, not a limitation unique to HERO. We are actively investigating this and plan to address it in future work.
>
> *Homogeneous graphs*: HERO is applicable—G-MM is architecture-agnostic; DiSCo degenerates to coverage-maximization + single-hop expansion (losing metapath benefits); HKD's L_sem extends via node-similarity attention (Appendix A.1, Eqs 14–17). HERO on homogeneous graphs is competitive with existing CGL methods, though its heterogeneous-specific advantages would not apply.

---

> > ### Author Rebuttal · Reviewer_mrVg · 2026-04-04
> >
> > I appreciate the authors' response. I have no further comment on W2/W3.
> >
> > For W1, while the response clarified the contribution to some extent, overall the contributions still seem incremental to me. Hence, I maintain my ratings.

---

> > > ### Author Response · Authors · 2026-04-04
> > >
> > > Dear reviewer,
> > >
> > > We sincerely thank you for your time and effort for engaging with our response and for confirming that the other two concerns have been resolved.
> > >
> > > We respectfully provide further clarification on why we believe HERO represents more than an incremental combination of existing ideas.
> > >
> > > First, regarding the problem itself: **HCGL is a new and previously unformalized problem**. Heterogeneous graphs introduce forgetting sources absent in homogeneous CGL, semantic imbalance across node types, uneven category arrival, and cross-type dependency shifts. HERO is the first framework to formally identify these challenges and design targeted solutions for them. This level contribution is independent of the individual techniques proposed.
> > >
> > > Second, regarding the framework design: HERO's three components are not independently plugged together. They form a well-integrated system where each component addresses a distinct forgetting source that the others cannot.
> > >
> > > Specifically:
> > >
> > > - **G-MM** improves plasticity, enabling the model to adapt quickly to heterogeneous task distributions. Its effect becomes more pronounced as heterogeneity increases—for example, from a 0.3% gain on DBLP (3 types) to 3.4% on Freebase (8 types) (Table 2). This trend suggests that the improvement is tied to heterogeneity rather than a generic meta-learning effect.
> > >
> > > - **DiSCo** addresses semantic imbalance by expanding subgraphs along metapaths (Eqs. 3–5). This type of sampling is not present in existing continual graph learning methods.
> > >
> > > - **HKD** introduces a distillation objective $\(L_{\text{sem}}\)$ (Eq. 10) that aligns metapath-level attention coefficients. This provides a signal tailored to heterogeneous graphs, helping preserve cross-type relational knowledge. The teacher model transfers this information to the student, which is initialized by G-MM, forming a coherent plasticity–stability mechanism.
> > >
> > > The ablation in Table 2 confirms this synergy: removing any single component degrades performance, and the degradation patterns differ across datasets precisely because each component targets a different heterogeneity-specific challenge. Appendix E (Fig 7) further shows that logit + semantic distillation outperforms either alone, confirming non-redundancy.
> > >
> > > Third, beyond the framework, the individual components contain algorithmic novelty: DiSCo's two-stage coverage-maximization + relation-type-aware expansion (Eqs 2–6) and HKD's metapath attention alignment (Eq 10) are new algorithms without direct precedent in CGL literature.
> > >
> > > We understand that subjective assessments of novelty may differ, and we respect the reviewer's perspective. We do want to emphasize that all other raised concerns (W2, W3) have been addressed with new experimental evidence, and we hope the reviewer will also consider the overall contribution holistically: the new problem formalization, the controlled experiments demonstrating each component's necessity, and the consistent improvements across 4 datasets and 3 HGNN backbones.
> > >
> > > We sincerely appreciate the reviewer's time and constructive feedback in the discussion process. We are committed to continuing to improve this work and will address remaining concerns in the revision.

---

### Official Review · Reviewer_FDVQ · 2026-03-11

**Soundness:** 2
**Presentation:** 3
**Significance:** 2
**Originality:** 2
**Overall Recommendation:** 4
**Confidence:** 4

**Summary:**

This paper studies continual learning on heterogeneous graphs, proposing HERO which combines gradient-based meta-learning for fast adaptation, diversity-aware sampling (DiSCo) for memory-efficient replay, and heterogeneity-aware knowledge distillation for knowledge retention. Experiments on four benchmarks show improvements over existing continual graph learning baselines.

**Compliance With Llm Reviewing Policy:**

Affirmed.

**Final Justification:**

The author's response effectively addressed my main concerns regarding baseline comparisons and ablation experiments, and I am inclined to improve the score.

**Key Questions For Authors:**

refer to weaknesses

**Strengths And Weaknesses:**

Strengths
1. The problem setting is relevant as heterogeneous graphs in practice evolve over time.
2. DiSCo sampling strategy that expands subgraphs along metapaths is sensible for preserving structural context.
3. The two-level distillation design adds depth to knowledge retention.

Weaknesses

1. Framework introduces significant complexity with three modules.
2. Existing continual graph learning methods required adaptation for fair comparison. I recommend incorporating more recent continual graph learning methods into the discussion and comparison, such as:

[1] Replay-and-Forget-Free Graph Class-Incremental Learning: A Task Profiling and Prompting Approach

[2] A Topology-aware Graph Coarsening Framework for Continual Graph Learning

[3] What Matters in Graph Class Incremental Learning? An Information Preservation Perspective

3. Meta-learning module contribution appears less critical in ablation study compared to replay and distillation.

---

> ### Author Rebuttal · Authors · 2026-03-30
>
> We thank Reviewer FDVQ for the insightful feedback. We address each concern below.
>
> **[W1 Framework Complexity]**
> The three modules address orthogonal, non-overlapping HCGL challenges: G-MM plasticity (fast adaptation via meta-learned initialization), DiSCo memory-efficient (with fewer stored nodes) heterogeneous replay, and HKD stability (forgetting prevention via multi-level distillation). Table 2 demonstrates that removing any single module degrades both AP and AF across all four datasets. Through this integrated system, we can effectively address HCGL, which cannot be achieved by any single module alone. In future work, we will explore more concise and effective approaches.
>
> **[W2 Missing Recent Baselines]**
> We appreciate these suggestions. However, all three methods operate in fundamentally incompatible settings:
>
> *Niu et al. 2024* ("Replay-and-Forget-Free"): Targets Graph Class-Incremental Learning (GCIL) with class-incremental splits and task-identity prompts on homogeneous graphs. Our setting is domain-incremental, where the task objective stays fixed while the heterogeneous graph structure expands. These settings require different forgetting strategies and the method has no mechanism for multiple node types or metapath semantics.
>
> *Han et al. 2024* ("Topology-aware Graph Coarsening"): Graph coarsening reduces graphs to smaller proxy structures. While effective for homogeneous settings, coarsening operations collapse node type distinctions, such as the Actor-Movie-Director metapath structure in IMDB, that cannot survive type-agnostic coarsening.
>
> *"What Matters in GCIL?"*: Also targets homogeneous GCIL, focusing on information preservation from a class-incremental perspective. The insights do not transfer directly to HCGL's heterogeneous-specific forgetting sources (semantic imbalance, cross-type dependency shifts).
>
> Adapting any of these to HCGL would require the same retrofitting applied to our current baselines. We will add detailed comparisons and citations for all three in the related work section of the revision.
>
> **[W3 G-MM Contribution]**
> G-MM's role is fast adaptation (plasticity), not forgetting prevention—the latter is handled by ER+HKD. Measuring G-MM solely via AP/AF metrics understates its contribution. Three lines of evidence:
>
> *(1) Heterogeneity-correlated contribution (Table 2)*: The AP drop from removing G-MM scales precisely with graph heterogeneity—0.3% on DBLP (4 node types), 1.2% on IMDB, 2.2% on Yelp, and 3.4% on Freebase (8 node types, 36 relation types). Higher semantic diversity creates larger inter-task distribution gaps where meta-learned initialization provides greater adaptation benefit.
>
> *(2) Plasticity metric—FaG (Table 8, Appendix G)*: The Forgetting-aware Gap captures performance degradation on the final task under the CL constraint. HERO's FaG is substantially lower than MetaCLGraph—9.90 vs. 13.68 on Freebase, 6.88 vs. 13.61 on Yelp. Since MetaCLGraph also uses meta-learning + replay (without HKD), this comparison isolates the G-MM + HKD synergy in preventing the final-task accuracy collapse that appears when HKD alone governs knowledge transfer.
>
> *(3) Shot sensitivity (Fig 4, Sec 5.4)*: G-MM is most critical at 5-shot settings where ER samples are insufficient to cover new task distributions. Meta-initialization enables fast generalization from few new examples, and performance peaks at 10-shot—the optimal balance between meta-generalization and task-specific adaptation.
>
> Overall, the G-MM module primarily supports the integrated system rather than being designed solely to improve performance; therefore, its full contribution cannot be adequately reflected by the AP metric alone.

---

> > ### Author Rebuttal · Reviewer_FDVQ · 2026-04-03
> >
> > The author's response effectively addressed my main concerns regarding baseline comparisons and ablation experiments, and I am inclined to improve the score.

---

> > > ### Author Response · Authors · 2026-04-04
> > >
> > > Dear reviewer,
> > >
> > > We would like to thank you for your time and valuable feedback. We are glad that our responses helped resolve the reviewer's concerns. If accepted, we adhere to using the one additional page in camera-ready version to address all the reviewers suggestions. We hope our work earns your support during the final review phase.
> > >
> > > Best regards,
> > > Authors

---

### Decision · Program_Chairs · 2026-04-30

**Decision:**

Reject

**Comment:**

This paper addresses the important problem of Heterogeneous Continual Graph Learning with a well-integrated framework and solid experimental validation. However, several concerns remain unresolved after the rebuttal. First, the fairness of baseline comparisons is questionable: all baselines use a simple sampling strategy for non-target nodes while HERO uses its own DiSCo sampling, which confounds the attribution of performance gains. A controlled experiment provided by the authors did not fully resolve this concern. Second, one reviewer continues to view the overall novelty as incremental. These unresolved issues regarding evaluation fairness and novelty outweigh the paper's strengths.